# UniLiP: Adapting CLIP for Unified Multimodal Understanding, Generation and Editing

**Hao Tang**[1,2], **Chenwei Xie**[2], **Xiaoyi Bao**[2,3], **Tingyu Weng**[2],
**Pandeng Li**[2], **Yun Zheng**[2], **Liwei Wang**[1,4,5]

[1]Center for Data Science, Peking University  [2]Alibaba Group
[3] CASIA  [4] Center for Machine Learning Research, Peking University
[5] State Key Laboratory of General Artificial Intelligence, Peking University, Beijing, China
{tanghao@stu, wanglw@cis}.pku.edu.cn  baoxiaoyi2021@ia.ac.cn
{eniac.xcw, wengtingyu.wty, lipandeng.lpd, zhengyun.zy}@alibaba-inc.com

## Abstract

In this paper, we propose UniLIP, a unified framework that adapts CLIP for multimodal understanding, generation and editing. Although CLIP excels at understanding, it lacks reconstruction abilities required to be a unified visual encoder. However, previous CLIP-based unified methods fail to balance understanding and reconstruction, leading to semantic degradation or inconsistent reconstructions. In contrast, we introduce a novel two-stage training scheme with a self-distillation strategy that progressively endows CLIP with high-fidelity reconstruction abilities while preserving its original comprehension performance. For enhanced reasoning and consistency in generation and editing, we further develop a dual-condition architecture built upon the MetaQuery framework. Our architecture jointly utilizes multimodal hidden states for rich contextual details and learnable query embeddings to harness the powerful reasoning abilities of Multimodal Large Language Models (MLLMs). Leveraging advanced image representation and architectural design, UniLIP demonstrates superior instruction following and edit fidelity. With only 1B and 3B parameters, UniLIP can outperform larger unified models such as BAGEL (7B) and Uniworld-V1 (12B), achieving state-of-the-art performance of **0.90** on GenEval, **0.63** on WISE, and **3.94** on ImgEdit. These results demonstrate that UniLIP successfully expands the application of CLIP, establishing its continuous features to not only serve as the optimal choice for understanding tasks but also achieve highly competitive performance in generation and editing tasks. Code and models are available at https://github.com/nnnth/UniLIP.

## 1 Introduction

The success of Large language models (LLMs) has driven unified modeling into multimodal domains, particularly in image understanding and generation. However, traditional approaches for these two tasks differ significantly: image understanding models align CLIP-like (Radford et al., 2021) semantic encoders with LLMs (Bai et al., 2025; Zhu et al., 2025), while image generation models either employ diffusion to model VAE latents (Xie et al., 2024a; Esser et al., 2024) or apply autoregressive modeling on discrete tokens encoded by VQVAE (Van Den Oord et al., 2017; Sun et al., 2024a). This divergence has sparked significant interest in a more unified approach.

Early efforts (Team, 2024; Zhou et al., 2024) combine VQVAE or VAE with LLMs for unification, but often achieve subpar performance in understanding because the encoder struggles to capture rich semantics. To mitigate this, recent works build unified tokenizers based on CLIP (Wu et al., 2024b; Sun et al., 2024b; Qu et al., 2024; Han et al., 2025). While CLIP features possess rich semantics, they lack pixel details, requiring extra modifications for reconstruction. As depicted in Figure 1 (a), VILA-U *et al.* (Wu et al., 2024b; Qu et al., 2024) discretize CLIP features, at the cost of information loss and worse understanding than the original CLIP. Emu2 (Sun et al., 2024b) freezes CLIP and finetunes a diffusion decoder to reconstruct images from CLIP features. However, as CLIP features lose pixel details, the diffusion decoder cannot produce consistent outputs (example in Figure 1),

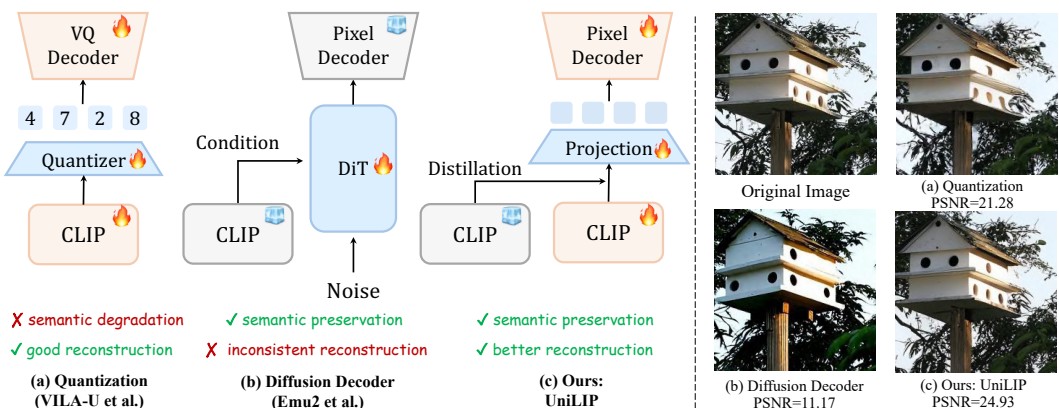

Figure 1: **CLIP-based reconstruction methods.** (a) Quantizing CLIP features into discrete tokens. (b) Using CLIP features as the condition for the diffusion transformer. (c) Ours: Training CLIP for reconstruction with self-distillation constraints. The right part shows a comparison of reconstruction results. The diffusion decoder produces incorrect hole positions and counts, while quantization-based methods exhibit distortions around the holes. In contrast, UniLIP achieves the best quality.

hindering editing tasks. Therefore, these methods either compromise CLIP's original understanding performance or achieve inconsistent reconstruction, leading to a trade-off in unification.

To build an optimal unified tokenizer based on CLIP, we identify two key challenges: (1) How to adapt CLIP for reconstruction without compromising its original comprehension capabilities? Training CLIP solely for reconstruction will lead to catastrophic forgetting in comprehension. (2) How to effectively utilize CLIP in generation and editing? While BLIP3-o (Chen et al., 2025a) validates CLIP is a better generation representation than VAE, it follows the reconstruction method of Emu2 (Sun et al., 2024b), requiring an extra diffusion process to decode CLIP features into images. This design not only adds computational costs but also hinders editing that demands consistency. UniWorld-V1 (Lin et al., 2025a) supports editing by conditioning on SigLIP (Zhai et al., 2023) features, but is limited to high-resolution images and depends on VAE features for generation, which are misaligned with the SigLIP features. Therefore, a more effective approach is required to enhance CLIP's reconstruction and establish it as a bridge between understanding and generation tasks.

In this paper, we propose UniLIP, adapting CLIP for reconstruction, generation and editing without losing original understanding capabilities. For reconstruction, we propose a two-stage training approach: the first stage trains only the pixel decoder to align with the frozen CLIP, and the second stage enables CLIP for training. To further restrict feature distribution changes, as shown in Figure 1 (c), we adopt a self-distillation loss in the second stage, using the frozen CLIP as the teacher. To enable generation with UniLIP features, we follow the architecture of MetaQuery (Pan et al., 2025). However, this approach is ineffective for editing since fixed-length queries fail to capture rich information in visual features, leading to inconsistencies. Therefore, we propose a dual-condition architecture: both the multimodal hidden states of the MLLM and the query embedding are used as conditions for diffusion. This approach not only fully utilizes the information in UniLIP features, but also leverages the reasoning capabilities of the MLLM through the query embedding.

We implement UniLIP based on InternViT from InternVL3 (Zhu et al., 2025). After our two-stage training, UniLIP achieves the best reconstruction performance among CLIP-based methods while preserving understanding performance. Surprisingly, UniLIP also achieves improvements on several comprehension benchmarks (see Table 1), likely because reconstruction training allows UniLIP to capture more image details (see Figure 2). For generation and editing, we train with 40M public images. As our unified representations are well-aligned with text and reconstructable, UniLIP can achieve superior prompt alignment and edit consistency. In image generation, UniLIP outperforms all similarly-sized unified models on the GenEval and WISE benchmarks, scoring **0.90** and **0.63** respectively. In image editing, UniLIP obtains a score of **3.94** on ImgEdit, significantly surpassing larger models like UniWorld-V1 (Lin et al., 2025a) and BAGEL (Deng et al., 2025).

In summary, our main contributions are listed as follows:

(1) We propose UniLIP, which inherits CLIP's state-of-the-art understanding capabilities and extends its application to generative tasks, achieving strong performance on multiple benchmarks.

(2) We introduce a novel training approach that extends CLIP for image reconstruction, overcoming its deficiency in capturing pixel details while preserving its original comprehension abilities.

(3) We propose a dual-condition architecture to bridge MLLM and the diffusion transformer, which effectively avoids information loss and maximizes UniLIP's potential in editing tasks.

| Model | Reconstruction | | | Understanding | | | | |
|---|---|---|---|---|---|---|---|---|
| | rFID↓ | PSNR↑ | SSIM↑ | MME-P↑ | MMBench↑ | MMVP↑ | AI2D↑ | TextVQA↑ |
| Frozen CLIP (Zhu et al., 2025) | 6.14 | 16.26 | 0.572 | 1492 | **72.6** | 67.3 | 69.4 | 74.1 |
| **UniLIP** | **0.31** | **24.62** | **0.788** | **1499** | **72.6** | **68.7** | **70.7** | **74.7** |

Table 1: **Reconstruction and understanding performance comparisons**. "Frozen CLIP" refers to the InternViT, and we train a pixel decoder to reconstruct images from its features.

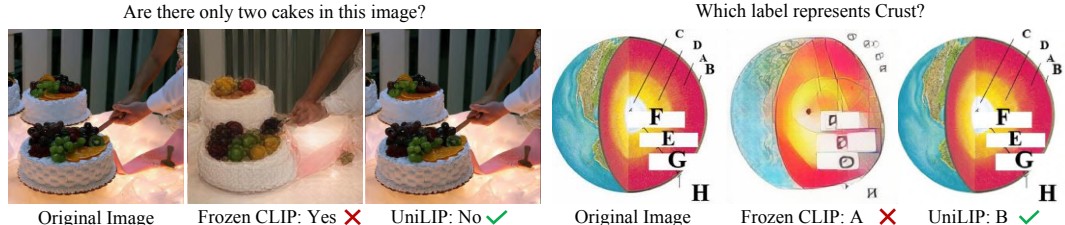

Figure 2: **Qualitative comparison of reconstruction and understanding results.** In the first example, the reconstructed image from frozen CLIP only shows two cakes, and the cake on the left edge is indistinguishable. In the second example, the letters are not visible for frozen CLIP. This lack of detail results in misunderstanding, while UniLIP can capture them and answer correctly.

## 2 RELATED WORK

**MLLMs for Image Understanding.** Initial approaches (Liu et al., 2023; Zhu et al., 2023; Dai et al., 2023) align the features of visual encoders to LLM input spaces, achieving strong performance through instruction tuning. In this architecture, the visual encoder is crucial for perception, and CLIP (Radford et al., 2021) has become the optimal choice due to its robust alignment with text. Subsequent methods have explored the use of better CLIP, including SigLIP (Zhai et al., 2023) and InternViT (Chen et al., 2024d), enhancing performance through improved designs (Shi et al., 2024; Tong et al., 2024a), larger model sizes (Liu et al., 2024a), better training strategies (Deitke et al., 2024), and extension to fine-grained perception tasks (Wang et al., 2024b; Tang et al., 2025). Nevertheless, these models remain specialized in comprehension without generative capabilities.

**Unified Visual Tokenizers.** Early works use VQ/VAE as unfied encoders (Zhou et al., 2024), which suffer from poor understanding due to limited semantics. Subsequent methods decouple the encoder, using CLIP for understanding and VQ/VAE for generation (Wu et al., 2024a), which improves performance but increases complexity. To simplify this, recent approaches build unified encoders upon CLIP. Some quantize CLIP features for reconstruction (Wu et al., 2024b; Qu et al., 2024; Ma et al., 2025a), causing semantic degradation. Other methods freeze CLIP and train a diffusion decoder to generate images conditioned on CLIP features (Sun et al., 2024b). However, since CLIP features lose image details, the reconstruction quality is poor. In contrast, UniLIP can maintain CLIP's understanding capability while achieving excellent reconstruction.

**Architectures for Unified MultiModal Understanding and Generation.** Early methods share weights between tasks (Team, 2024), leading to training instability. Later methods adopt Mixture-of-Experts (MoE) to decouple tasks, resolving conflicts but doubling parameters (Liao et al., 2025; Deng et al., 2025). To improve integration, the pioneering work DreamLLM (Dong et al., 2023) first proposes learnable queries to bridge MLLMs and diffusion models, facilitating synergistic joint training. MetaQuery (Pan et al., 2025) further extends this to connect frozen MLLMs with DiTs. BLIP3-o (Chen et al., 2025a) adopts a similar framework but uses CLIP features as diffusion targets for better prompt alignment. While UniLIP also uses learnable queries, we introduce a dual-condition design to address the inability of prior query-based methods in editing.

**Instruction-Guided Image Editing.** Early methods (Brooks et al., 2023; Zhang et al., 2023) employ modular pipelines, using MLLMs to generate prompts or spatial cues that guide editing. To enhance semantic control, many approaches (Liu et al., 2025; Wu et al., 2025a) utilize MLLMs to extract rich semantic embeddings from both image and text, thereby conditioning the generative backbone for more faithful and controllable edits. However, since CLIP in MLLMs cannot capture pixel details, these methods still require a VAE for supplementary. UniWorld-V1 (Lin et al., 2025a) finds that SigLIP can preserve consistency at high resolution, but still relies on VAE features for generation. In contrast, UniLIP can achieve consistent editing by adapting CLIP for reconstruction, maintaining coherence without resolution constraints and VAE dependence.

## 3 METHOD

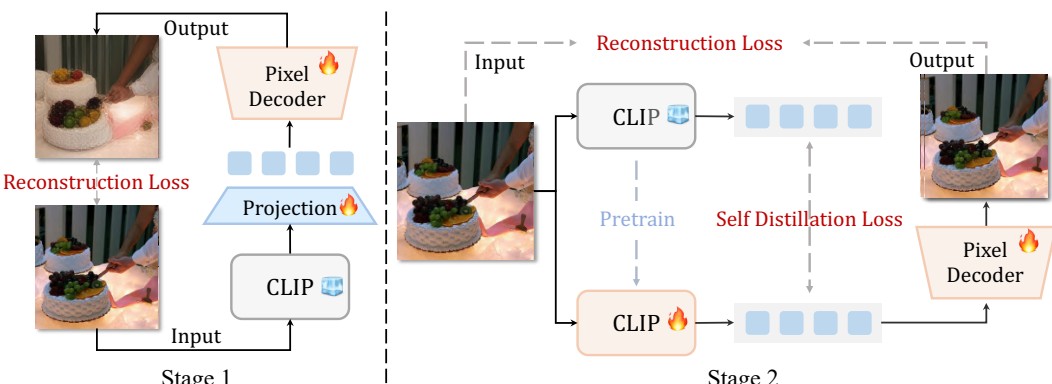

Figure 3: **Reconstruction training of UniLIP.** Stage 1 trains the pixel decoder to learn reconstruction based on frozen CLIP features, only producing blurry reconstruction results. Stage 2 enables CLIP training while incorporating self-distillation supervision to maintain the original capability. The projection between CLIP and the pixel decoder is omitted for simplicity.

### 3.1 FINETUNING CLIP FOR IMAGE RECONSTRUCTION

Image reconstruction demands fine-grained details, whereas CLIP primarily encodes high-level semantics. Therefore, the central challenge is to inject details without eroding original semantics. In pilot experiments, we reconstruct images directly from frozen CLIP features. The output is noticeably blurry (Figure 3, Stage 1), yet revealing that CLIP still retains very weak pixel-level cues. Guided by this insight, we design a two-stage training scheme to locate and then amplify CLIP's latent reconstruction capacity. During the second stage, we introduce a self-distillation loss to constrain distributional drift, thereby boosting reconstruction without impairing the original capacity.

**Architecture.** As illustrated in Figure 3, we employ an autoencoder architecture by pairing CLIP with a pixel decoder $D_{\text{pix}}$, using a projection $h_\phi$ to align feature dimensions. The reconstruction process is:
$$\hat{I} = D_{\text{pix}}(h_\phi(\text{CLIP}(I)))$$
Specifically, the input image $I \in \mathbb{R}^{H \times W \times 3}$ is processed by CLIP to extract features $F_{\text{clip}} \in \mathbb{R}^{\frac{H}{p} \times \frac{W}{p} \times d}$, where $p$ is the downsample ratio and $d$ is the feature dimension of CLIP. Then $h_\phi$ aligns the features to the input dimension of $D_{\text{pix}}$, obtaining $F_{\text{proj}} \in \mathbb{R}^{\frac{H}{p} \times \frac{W}{p} \times \hat{d}}$. Finally, the pixel decoder $D_{\text{pix}}$ reconstructs $\hat{I}$ from $F_{\text{proj}}$. In practice, $h_\phi$ is implemented using several MLPs.

**Stage 1: Aligning the Pixel Decoder with CLIP.** In this stage, we fix the CLIP and train the pixel decoder $D_{\text{pix}}$ along with the projection $h_\phi$. This configuration allows the model to fully leverage the information embedded within CLIP features while preventing unnecessary alterations. To accelerate the convergence of training, the pixel decoder is initialized with pre-trained weights, while the projection is randomly initialized. The training objective is defined as:
$$\mathcal{L}_{\text{stage1}} = \mathcal{L}_{\text{MSE}} + \mathcal{L}_{\text{LPIPS}}$$
where $\mathcal{L}_{\text{MSE}}$ represents pixel-wise reconstruction loss, and $\mathcal{L}_{\text{LPIPS}}$ represents the perceptual loss computed using the LPIPS metric (Zhang et al., 2018).

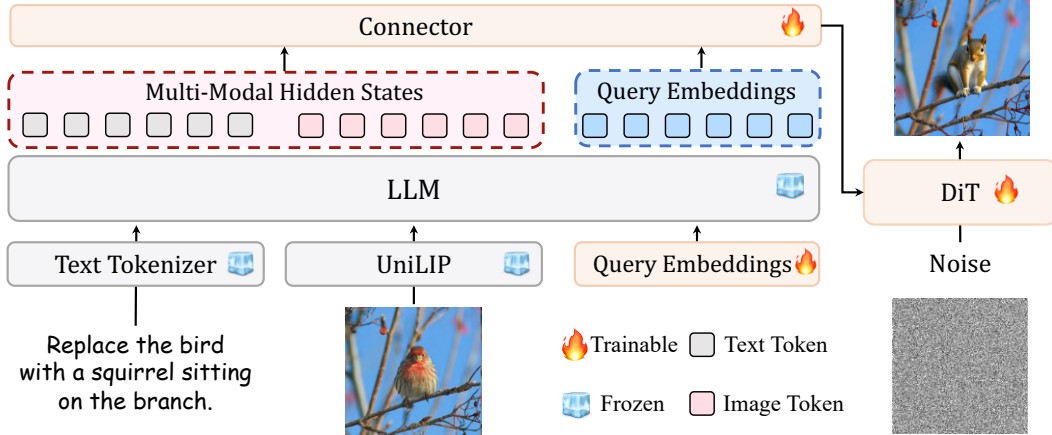

Figure 4: **Overview of the dual-condition architecture in editing tasks.** We replace InternViT with our UniLIP and freeze LLM to keep the original understanding performance. When generating images, UniLIP concatenates the multimodal hidden states and query embeddings as the input to the connector, which then serves as the condition for the diffusion transformer. This architecture effectively avoids information loss while leveraging the reasoning capabilities of LLMs through query embeddings. For generation tasks, simply remove the input of the reference image.

**Stage 2: Refining CLIP with Self-Distillation.** Since CLIP features inherently lack fine-grained pixel details, the reconstruction quality achieved in the first stage is unsatisfactory. Therefore, in the second stage, we allow CLIP to undergo training. To prevent the features from rapidly diverging from their original distribution, we employ a self-distillation approach to constrain feature alterations. The training objective in this stage is:

$$\mathcal{L}_{\text{stage2}} = \mathcal{L}_{\text{MSE}} + \mathcal{L}_{\text{LPIPS}} + \lambda \|F_{\text{orig}} - F_{\text{ft}}\|_2^2$$

where $\lambda$ is the weight for the distillation loss, $F_{\text{orig}}$ denotes the original CLIP features, and $F_{\text{ft}}$ is the fine-tuned CLIP features. We empirically find that setting $\lambda$ to 1 is sufficient (see Table 11). Additionally, to further constrain parameter updates in CLIP, we set its learning rate to 0.1 times the global learning rate. We find that this strategy also helps preserve original abilities (Table 6).

Through our reconstruction training, we obtain a stronger UniLIP that not only supports reconstruction but also retains its original understanding performance, even achieving higher scores on several benchmarks (see Table 1). These results indicate that we have partially overcome CLIP's prior shortcomings in capturing image details. UniLIP is also well-suited to generation and editing: (1) UniLIP can achieve a $32\times$ compression ratio relative to the original image while remaining directly decodable into pixels by a lightweight decoder. (2) UniLIP exhibits strong alignment with text, leading to better prompt consistency in generation and a clearer correspondence between textual instructions and visual inputs during editing; (3) UniLIP can encode both pixel-level details and high-level semantics, thereby supplying all the information of the reference image for reliable editing.

## 3.2 DUAL CONDITIONS FOR IMAGE GENERATION AND EDITING

As UniLIP produces continuous features, we follow methods like DreamLLM (Dong et al., 2023) and MetaQuery (Pan et al., 2025) to build the generation pipeline. These methods employ query embeddings to bridge MLLM with a diffusion transformer, where the query tokens serve as the conditions for the generation process. Notably, in knowledge-augmented generation, query embeddings encode the MLLM's reasoning result to implicitly define the target image content, resulting in better prompt alignment. However, such methods are rarely applied to editing tasks and have not demonstrated satisfactory quality or robust quantitative metrics in editing benchmarks.

We identify the main bottleneck as the fixed number of query embeddings. For example, DreamLLM uses 64 tokens and MetaQuery uses 256 tokens, which limit the expressive capacity of the queries. This problem is not obvious in text-to-image generation, since most text prompts in generation benchmarks are short and LLMs excel at compressing text information. However, in editing, query embeddings need to retain the details of one or more reference images. Therefore, using a fixed number of tokens is inevitable for information loss, which leads to inconsistency in the editing.

| Model | Res. | ratio | rFID ↓ | PSNR↑ | SSIM↑ |
|-------|------|-------|--------|-------|-------|
| VILA-U (Wu et al., 2024b) | 256 | 16 | 1.80 | - | - |
| Tokenflow (Qu et al., 2024) | 256 | 16 | 1.37 | 21.41 | 0.687 |
| DualViTok (Huang et al., 2025) | 256 | 16 | 1.37 | 22.53 | 0.741 |
| **UniLIP** | 256 | 32 | **0.79** | **22.99** | **0.747** |
| Emu2 (Sun et al., 2024b) | 448 | 14 | 3.27 | 13.49 | 0.423 |
| **UniLIP** | 448 | 32 | **0.31** | **24.62** | **0.788** |

Table 2: **CLIP-based reconstruction performance.**

To address this information bottleneck, we propose a dual-condition architecture to connect the MLLM and the diffusion transformer. As shown in Figure 4, in addition to the query embeddings, the multimodal hidden states of the MLLM are also used as conditions, forming dual conditions for cross attention in DiT. This supplements the information that cannot be captured by the query embeddings alone, such as pixel details from the reference images.

Our architecture successfully decouples generation and editing into complementary subtasks: the MLLM extracts rich contextual information and performs reasoning to determine the outcome, while the DiT synthesizes the image conditioned on the extracted and inferred cues. Critically, the proposed dual-condition framework guarantees lossless information propagation during this decoupling, thereby allowing UniLIP to fully exploit its advantages in both generation and editing.

### 3.3 TRAINING SCHEMES

To construct a unified model based on UniLIP capable of simultaneous understanding, generation, and editing, we adopt a three-stage training strategy:

**Stage 1: Connector Training.** The primary goal of this stage is to train the connector to effectively bridge the MLLM and the DiT. The connector learns to align the MLLM's output features with the DiT's original conditional feature space. Parameters of both the MLLM and the DiT are frozen in this stage, and training is conducted exclusively on generation tasks.

**Stage 2: Unified Generation and Editing Training.** We utilize large-scale datasets to train the model on general generation and editing. Both the connector and the DiT are trained in this stage. Consistent with prior methods (Chen et al., 2025a), the MLLM remains frozen, eliminating the need for expensive training on understanding tasks.

**Stage 3: Supervised Fine-Tuning (SFT).** We employ high-quality instruction tuning data to enhance generation fidelity and prompt alignment for both generation and editing tasks. During this phase, the connector and DiT continue to be optimized, while the MLLM remains frozen.

## 4 EXPERIMENTS

### 4.1 IMPLEMENTATION DETAILS

**Architectures.** We introduce two model variants, UniLIP-1B and UniLIP-3B, which are constructed by integrating the InternVL3 (Zhu et al., 2025) with the SANA (Xie et al., 2024a). Specifically, UniLIP-1B combines InternVL3-1B with SANA-0.6B, whereas UniLIP-3B utilizes InternVL3-2B and SANA-1.6B. We directly employ the InternViT from InternVL3 as the CLIP and adopt the pixel decoder from DC-AE (Chen et al., 2024b). Our connector consists of 6 layers, maintaining the same structure and dimensions as the LLM in InternVL3. For the learnable queries, we set $N$=256.

**Training Data.** We use the image generation data from BLIP3-o (Chen et al., 2025a). The pretraining data includes 27M samples recaptioned by Qwen2.5-VL-7B (Bai et al., 2025), 5M samples from CC12M (Changpinyo et al., 2021) and 4M synthesized images from JourneyDB (Sun et al., 2023). The instruction tuning data consists of 60K high-quality image-text pairs, generated by GPT-4o (OpenAI, 2025). For editing, we use data from GPT-Image-Edit-1.5M (Wang et al., 2025) for pretraining. In instruction tuning, we use ShareGPT-4o-Image (Chen et al., 2025b), which contains 46K editing samples. Since we freeze the LLM, understanding data is not required.

**Image Reconstruction Training.** We train on the BLIP3-o pretraining data using 4 A100 GPUs with a batch size of 64 and a learning rate of 1e-4. The first stage runs for 300k steps at 224× 224

| Model | # LLM Params | MME-P | MMB | MMMU | MM-Vet | SEED | AI2D | MMVP |
|---|---|---|---|---|---|---|---|---|
| *Und. Only* | | | | | | | | |
| LLaVA-OV (Li et al., 2024a) | 1B | 1238 | 52.1 | 31.4 | 29.1 | 65.5 | 57.1 | - |
| InternVL2.5 (Chen et al., 2024c) | 1B | - | 70.7 | 41.2 | 48.8 | - | 69.3 | 31.3 |
| InternVL3 (Zhu et al., 2025) | 1B | 1492 | 72.6 | 43.4 | 59.5 | 71.1 | 69.4 | 67.3 |
| InternVL2.5 (Chen et al., 2024c) | 1.8B | - | 74.7 | 43.6 | 60.8 | - | 74.9 | - |
| InternVL3 (Zhu et al., 2025) | 1.8B | 1633 | 80.6 | 48.2 | 62.2 | 75.0 | 78.5 | 72.7 |
| Qwen2.5-VL (Bai et al., 2025) | 3B | - | 79.1 | 53.1 | 61.8 | - | 81.6 | - |
| Emu3-Chat (Wang et al., 2024c) | 8B | 1244 | 58.5 | 31.6 | 37.2 | 68.2 | 70.0 | 36.6 |
| *Und. and Gen.* | | | | | | | | |
| Chameleon (Team, 2024) | 7B | - | 35.7 | 28.4 | 8.3 | - | - | 0.0 |
| VILA-U (Wu et al., 2024b) | 7B | 1336 | 66.6 | 32.2 | 27.7 | 56.3 | - | 22.0 |
| MetaMorph (Tong et al., 2024b) | 8B | - | 75.2 | 41.8 | - | - | - | 48.3 |
| SEED-X (Ge et al., 2024) | 13B | 1457 | 70.1 | 35.6 | 43.0 | 66.5 | - | - |
| TokenFlow-B (Qu et al., 2024) | 13B | 1354 | 55.3 | 34.2 | 22.4 | 60.4 | 54.2 | - |
| Show-O (Xie et al., 2024b) | 1.3B | 1097 | - | 26.7 | - | - | - | - |
| ILLUME (Wang et al., 2024a) | 7B | 1445 | 75.1 | 38.2 | 37.0 | - | 71.4 | - |
| Janus-Pro (Chen et al., 2025c) | 7B | 1567 | 79.2 | 41.0 | 50.0 | 72.1 | - | - |
| Harmon (Wu et al., 2025c) | 1.5B | 1155 | 65.5 | 38.9 | - | 67.1 | - | |
| MetaQuery-B (Pan et al., 2025) | 1B | 1238 | 58.5 | 31.4 | 29.1 | 66.6 | - | - |
| BAGEL (Deng et al., 2025) | 3B | 1610 | 79.2 | 43.2 | 48.2 | - | - | 54.7 |
| BLIP3-o (Chen et al., 2025a) | 4B | 1528 | 78.6 | 46.6 | 60.1 | 73.8 | - | - |
| TokLIP (Lin et al., 2025b) | 7B | 1410 | - | 42.1 | - | 65.2 | - | - |
| Tar (Han et al., 2025) | 7B | 1571 | 74.4 | 39.0 | - | 73.0 | - | - |
| **UniLIP-1B** | 1B | 1499 | 72.6 | 43.3 | 59.4 | 71.0 | 70.7 | 68.7 |
| **UniLIP-3B** | 2B | **1636** | **80.7** | **48.7** | **62.2** | **75.0** | **78.6** | **73.0** |

Table 3: **Comparison with state-of-the-arts on visual understanding benchmarks.**

| Model | # Params | GenEval | | | WISE | | |
|---|---|---|---|---|---|---|---|
| | | Counting | Position | Overall | Cultural | Biology | Overall |
| *Gen. Only* | | | | | | | |
| SDXL (Podell et al., 2023) | 2.6B | 0.39 | 0.15 | 0.55 | 0.43 | 0.44 | 0.43 |
| FLUX.1-dev (Labs, 2023) | 12B | 0.75 | 0.68 | 0.82 | 0.48 | 0.42 | 0.50 |
| PixArt-$\alpha$ (Chen et al., 2024a) | 0.6B | 0.44 | 0.08 | 0.48 | 0.45 | 0.49 | 0.47 |
| Emu3-Gen (Wang et al., 2024c) | 8B | 0.34 | 0.17 | 0.54 | 0.34 | 0.41 | 0.39 |
| SD3-Medium (Esser et al., 2024) | 2B | 0.72 | 0.33 | 0.74 | 0.42 | 0.39 | 0.42 |
| Sana-1.6B (Xie et al., 2024a) | 1.6B | 0.62 | 0.21 | 0.66 | - | - | |
| *Und. and Gen.* | | | | | | | |
| VILA-U (Wu et al., 2024b) | 7B | - | - | - | 0.26 | 0.35 | 0.31 |
| TokenFlow-XL (Qu et al., 2024) | 14B | 0.41 | 0.16 | 0.55 | - | - | - |
| ILLUME+ (Huang et al., 2025) | 3B + 2.6B | 0.62 | 0.42 | 0.72 | - | - | - |
| Janus-Pro (Chen et al., 2025c) | 7B | 0.59 | 0.79 | 0.80 | 0.30 | 0.36 | 0.35 |
| MetaQuery-B (Pan et al., 2025) | 1B + 1.6B | - | - | 0.74 | 0.44 | 0.41 | 0.46 |
| MetaQuery-XL (Pan et al., 2025) | 7B + 1.6B | - | - | 0.80 | 0.56 | 0.49 | 0.55 |
| Harmon (Wu et al., 2025c) | 1.5B + 1B | 0.66 | 0.74 | 0.76 | 0.38 | 0.37 | 0.41 |
| BLIP3-o-4B (Chen et al., 2025a) | 3B + 1.4B | - | - | 0.81 | - | - | 0.50 |
| BLIP3-o-8B (Chen et al., 2025a) | 7B + 1.4B | - | - | 0.84 | - | - | 0.62 |
| BAGEL (Deng et al., 2025) | 7B + 7B | 0.81 | 0.64 | 0.82 | 0.44 | 0.44 | 0.52 |
| OpenUni-B (Wu et al., 2025b) | 1B + 0.6B | 0.74 | 0.77 | 0.84 | 0.37 | 0.39 | 0.43 |
| OpenUni-L (Wu et al., 2025b) | 2B + 1.6B | 0.77 | 0.75 | 0.85 | 0.51 | 0.48 | 0.52 |
| Show-o2 (Xie et al., 2025) | 7B | 0.58 | 0.52 | 0.76 | 0.33 | 0.39 | 0.39 |
| Tar (Han et al., 2025) | 7B | 0.83 | 0.80 | 0.84 | - | - | - |
| **UniLIP-1B** | 1B + 0.6B | 0.83 | 0.83 | 0.88 | 0.54 | 0.50 | 0.56 |
| **UniLIP-3B** | 2B + 1.6B | **0.84** | **0.86** | **0.90** | **0.66** | **0.60** | **0.63** |

Table 4: **Evaluation of text-to-image generation ability on GenEval and WISE benchmark.**

resolution. The second stage consists of 400k steps at $224 \times 224$ followed by 100k steps at $448 \times 448$. We enable adversarial training (Karras et al., 2019) after 200k steps in the second stage. To further restrict parameter updates, the learning rate of CLIP is set to 1e-5 in the second stage.

**Image Generation and Editing Training.** Stages 1, 2, and 3 are trained for 50k, 200k, and 30k steps, respectively. All stages use a batch size of 512 with a cosine learning rate decay schedule ranging from 1e-4 to 1e-5.

| Model | # Params | Add | Adjust | Replace | Remove | Bkg. | Style | Overall |
|-------|----------|-----|--------|---------|--------|------|-------|---------|
| GPT-4o (OpenAI, 2025) | - | 4.61 | 4.33 | 4.35 | 3.66 | 4.57 | 4.93 | 4.20 |
| MagicBrush (Zhang et al., 2023) | 0.9B | 2.84 | 1.58 | 1.97 | 1.58 | 1.75 | 2.38 | 1.90 |
| Instruct-P2P (Brooks et al., 2023) | 0.9B | 2.45 | 1.83 | 2.01 | 1.50 | 1.44 | 3.55 | 1.88 |
| AnyEdit (Yu et al., 2025) | 1.3B | 3.18 | 2.95 | 2.47 | 2.23 | 2.24 | 2.85 | 2.45 |
| UltraEdit (Zhao et al., 2024) | 2.0B | 3.44 | 2.81 | 2.96 | 1.45 | 2.83 | 3.76 | 2.70 |
| OmniGen (Xiao et al., 2025) | 3.8B | 3.47 | 3.04 | 2.94 | 2.43 | 3.21 | 4.19 | 2.96 |
| Step1X-Edit (Liu et al., 2025) | 7B+12B | 3.88 | 3.14 | 3.40 | 2.41 | 3.16 | 4.63 | 3.06 |
| ICEdit (Zhang et al., 2025) | 12B | 3.58 | 3.39 | 3.15 | 2.93 | 3.08 | 3.84 | 3.05 |
| BAGEL (Deng et al., 2025) | 7B+7B | 3.56 | 3.31 | 3.30 | 2.62 | 3.24 | 4.49 | 3.20 |
| UniWorld-V1 (Lin et al., 2025a) | 7B+12B | 3.82 | 3.64 | 3.47 | 3.24 | 2.99 | 4.21 | 3.26 |
| Janus-4o (Chen et al., 2025b) | 7B | 3.60 | 3.25 | 3.27 | 2.28 | 3.32 | 4.47 | 3.26 |
| OmniGen2 (Wu et al., 2025a) | 3B+4B | 3.57 | 3.06 | 3.74 | 3.20 | 3.57 | 4.81 | 3.44 |
| **UniLIP-1B** | 1B+0.6B | 4.11 | 3.58 | 4.30 | 3.97 | 4.00 | **4.87** | 3.81 |
| **UniLIP-3B** | 2B+1.6B | **4.29** | **3.90** | **4.44** | **4.10** | **4.14** | 4.80 | **3.94** |

Table 5: **Evaluation of image editing ability on ImgEdit benchmark.**

## 4.2 COMPARISON WITH STATE-OF-THE-ARTS

**Image Reconstruction.** Table 2 compares the performance of CLIP-based tokenizers on the ImageNet 50k validation set. Since our 1B and 3B models employ vision encoders and pixel decoders with identical model size, their reconstruction performance is also nearly identical (see Appendix B.1). For brevity, we only report the results for UniLIP in 1B model. At a 256×256 resolution, UniLIP outperforms previous quantized CLIP tokenizers, achieving a 0.58 rFID improvement over DualViTok (Huang et al., 2025). Notably, UniLIP has a higher downsampling ratio, which enhances the efficiency of subsequent generation tasks. At 448×448 resolution, we benchmark against Emu2 (Sun et al., 2024b), which uses a diffusion decoder to reconstruct 1024×1024 images from 448×448 inputs. For a fair comparison, we resize Emu2's outputs to 448×448. Thanks to our dedicated reconstruction training, UniLIP significantly outperforms Emu2 with a 2.96 improvement in rFID and an 11.13 gain in PSNR. These results suggest that methods relying on diffusion decoders to supplement details often struggle to maintain fidelity to the original image. In contrast, our approach more effectively preserves and reconstructs pixel-level details from the CLIP features.

**Multimodal Understanding.** We replace InternViT in InternVL3 with our UniLIP for evaluation. The benchmarks include MME-P (MME-Perception) (Fu et al., 2023), MMB (MMBench) (Liu et al., 2024c), MMMU (Yue et al., 2024), MM-Vet (Yu et al., 2023), SEED (SEED-Bench Img) (Li et al., 2024b), AI2D (Kembhavi et al., 2016), and MMVP (Tong et al., 2024c). As shown in Table 3, we compare UniLIP with both unified models and understanding-only models (indicated as Und. Only). Because our reconstruction training can effectively maintain the original capabilities, UniLIP can achieve state-of-the-art understanding performance compared to previous unified models with similar scales. It is noteworthy that UniLIP's understanding performance significantly surpasses previous methods that quantize CLIP features, such as Tar (7B), VILA-U (7B), and TokenFlow (14B), even though our model size is much smaller.

**Image Generation.** We primarily evaluate the model's image generation capabilities on two benchmarks: GenEval (Ghosh et al., 2023) and WISE (Niu et al., 2025). GenEval examines how well the attributes of generated objects can be controlled by user instructions, including counting, position, and color attributes. WISE tests whether the model can understand text prompts that contain complex semantics and world knowledge. Table 4 presents the performance on GenEval and WISE benchmarks. Benefiting from our rich semantic features, the generation results of UniLIP align more effectively with text prompts, earning a score of 0.90 on the GenEval benchmark and 0.63 on the WISE benchmark. This result not only significantly outperforms models of similar size, such as Harmon, Janus-Pro, and MetaQuery, but is also comparable to larger models like BLIP3-o and BAGEL, indicating that UniLIP is a better choice for building a unified model based on MLLMs. Qualitative results are presented in Figure 5 and Figure 10 (comparisons), which showcase UniLIP's ability to generate aesthetically pleasing images that closely adhere to user prompts.

**Image Editing.** We primarily evaluate image editing capabilities on ImgEdit-Bench (Ye et al., 2025). As shown in Table 5, we achieve a highly competitive performance, with an overall score of 3.94, surpassing OmniGen2 (3.44) and UniWorld-V1 (3.26). This strong performance stems from the advantages of UniLIP's features, which are not only semantically rich and well-aligned with editing instructions but also retain fine-grained visual details. Our proposed dual-condition architec-

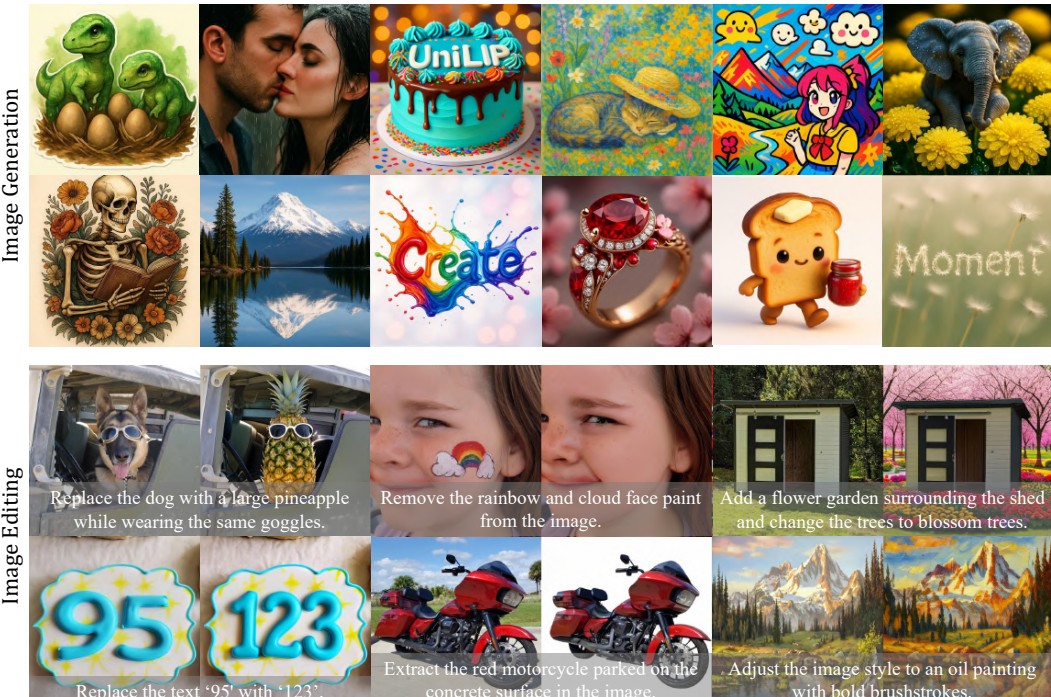

Figure 5: **Qualitative results of image generation and editing.**

ture is designed to fully leverage this rich information, enabling precise edits while preventing detail loss and maintaining consistency in unedited regions. The qualitative results in Figure 5 showcase UniLIP's ability to accurately modify images while preserving the integrity of surrounding areas. Further visual comparisons are provided in the appendix in Figure 11.

## 4.3 ABLATION STUDY

**Strategies in Reconstruction Training.** We present an ablation of the core design choices in reconstruction training in Table 6. The first row shows the baseline where CLIP is directly finetuned with the reconstruction objective. Although this achieves the best reconstruction PSNR, the understanding performance drops significantly, with results on several benchmarks falling to zero. All three training strategies we adopt help preserve the understanding ability of the CLIP, with only a minor loss in reconstruction performance compared to the baseline. Among them, self-distillation training proves to be the most effective; removing it leads to a 54.2 point decrease in MMBench.

**Two stage vs. Single stage.** To verify the effectiveness of the two-stage training strategy, we compare the training loss and performance of our approach against a single-stage baseline in Figure 6. We observe that the single-stage method exhibits severe instability, characterized by an initial distillation loss spike nearly double that of our method (0.0939 vs. 0.0497). Consequently, its convergence is significantly slower, requiring $3\times$ more iterations to reduce the loss and $4\times$ more to recover understanding performance. These results suggest that Stage 1 is critical for pre-aligning the pixel decoder; without this preliminary step, the initial misalignment between the unfrozen CLIP and the randomly initialized projection layer leads to gradient instability.

**Dual-condition Architecture.** Table 7 presents an ablation of the dual-condition architecture in UniLIP-1B. On the WISE benchmark, which requires knowledge-driven generation, query embeddings make better use of the MLLM's reasoning ability and lead to a 5-point improvement compared to using only the multimodal hidden states. In editing, however, relying solely on query embeddings results in poor performance, achieving a score of only 3.38 on ImgEdit, which is lower than using just the multimodal hidden states. This is primarily because the query embeddings cannot effectively compress the information in the reference image, resulting in inconsistency. The dual condition architecture combines the advantages of both approaches, thereby achieving optimal performance.

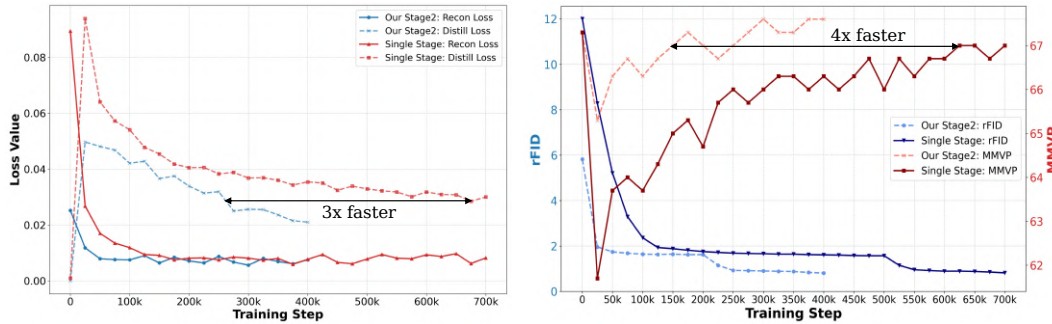

(a) Reconstruction and distillation loss curve. (b) Recon. (rFID) and und. (MMVP) performance

Figure 6: **Comparison of training loss and performance curve between single-stage and two-stage method**. Thanks to the pre-alignment in Stage 1, our Stage 2 can converge much faster.

| Two Stage | Self-Distillation | Lr Decay | Reconstruction | | | Understanding | | | | |
|---|---|---|---|---|---|---|---|---|---|---|
| | | | rFID↓ | PSNR↑ | SSIM↑ | MME-P↑ | MMBench↑ | MMVP↑ | AI2D↑ | TextVQA↑ |
| ✗ | ✗ | ✗ | 0.43 | 26.01 | 0.819 | 124 | 0 | 47.3 | 27.9 | 0 |
| ✓ | ✗ | ✓ | 0.29 | 25.28 | 0.804 | 709 | 18.4 | 50.0 | 50.0 | 7.5 |
| ✓ | ✓ | ✗ | 0.28 | 25.11 | 0.801 | 1466 | 71.4 | 66.7 | 68.3 | 67.3 |
| ✗ | ✓ | ✓ | 0.35 | 24.61 | 0.782 | 1478 | 72.0 | 67.3 | 69.5 | 74.0 |
| ✓ | ✓ | ✓ | 0.31 | 24.62 | 0.788 | 1499 | 72.6 | 68.7 | 70.7 | 74.7 |

Table 6: **Ablation of reconstruction training strategies on reconstruction and understanding performance.** "Lr Decay" refers to setting the learning rate of CLIP to 1e-5.

**Encoders for reference and target images.** Table 8 presents an ablation on encoder choices for reference and target images in UniLIP-1B. The target image encoder cannot be a CLIP, because it either produces blurry reconstructions or requires an additional DiT to generate images. Replacing the reference image encoder with CLIP (original InternViT) significantly decreases editing performance (from 3.81 to 3.42), as CLIP lacks pixel details, leading to mismatches between generated and reference images (see Appendix Figure 12). Using VAE (DC-AE) as the target image encoder also reduces generation and editing quality, especially on WISE (from 0.56 to 0.48). These findings indicate that UniLIP achieves superior prompt alignment compared to VAE, which is consistent with the results in BLIP3-o (Chen et al., 2025a). Overall, UniLIP serves as a more capable unified encoder than CLIP and VAE, offering richer pixel detail and enhanced instruction adherence.

## 5 CONCLUSION

In this paper, we propose UniLIP, a general framework that extends CLIP to support understanding, reconstruction, generation, and editing simultaneously, paving the way for more powerful and unified multimodal models. By introducing a carefully designed two-stage training scheme with a self-distillation constraint, UniLIP effectively overcomes the conventional trade-off between semantic comprehension and pixel-level detail preservation. Furthermore, the dual-condition architecture seamlessly bridges MLLMs and generative diffusion models, enabling image synthesis and editing with superior detail and consistency. We demonstrate the superiority of UniLIP across a wide range of benchmarks, setting a new promising direction for unified multimodal foundation models.

| Multimodal Hidden States | Query Embedding | WISE | ImgEdit |
|---|---|---|---|
| ✓ | ✗ | 0.47 | 3.62 |
| ✗ | ✓ | 0.52 | 3.38 |
| ✓ | ✓ | 0.56 | 3.81 |

Table 7: **Ablation of dual condition** architecture on WISE and ImgEdit benchmark.

| Reference | Target | GenEval | WISE | ImgEdit |
|---|---|---|---|---|
| CLIP | VAE | 0.84 | 0.46 | 3.37 |
| UniLIP | VAE | 0.85 | 0.48 | 3.70 |
| CLIP | UniLIP | 0.88 | 0.53 | 3.42 |
| UniLIP | UniLIP | 0.88 | 0.56 | 3.81 |

Table 8: **Ablation of image tokenizers** for reference (editing only) and target images in training.

## REPRODUCIBILITY STATEMENT

We have made every effort to ensure that the results presented in this paper are reproducible. The model configurations, datasets, hardware types, training and inference hyperparameters are described in detail in Section 4.1 and Appendix A. Moreover, the pre-trained models and datasets we utilized are all open-sourced and easily accessible.

## ACKNOWLEDGMENTS

LW is supported by National Science and Technology Major Project (2022ZD0114902) and National Science Foundation of China (NSFC92470123, NSFC62276005).

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

APPENDIX

# A    IMPLEMENTATION DETAILS

## A.1    ARCHITECTURE

To enhance the alignment between image features and the LLM, the UniLIP architecture incorporates both the visual encoder (CLIP) and the projection layers between the CLIP and the LLM (e.g., the two-layer MLP in InternVL). Images are downsampled $28\times$ ($14\times$ via CLIP, then $2\times$ via pixel shuffle). Since a $28\times$ upsampling decoder is unavailable, we use the closest option, a $32\times$ decoder (DC-AE-f32c32), and interpolate the output to the original resolution. A three-layer MLP then bridges the dimensionality gap, projecting UniLIP's high-dimensional features (e.g., 896 in 1B) down to the 32 dimensions accepted by the pixel decoder.

## A.2    TASK BALANCE

Since the amount of generative data is much larger than editing, we adjust the sampling ratio during joint training by setting the sampling proportion of editing data to ten times its data ratio. In this way, the sampling ratio of generative to editing data during training is approximately 2:1. During instruction tuning, we set the sampling proportion of editing data to three times its data ratio, making the sampling ratio of generation to editing data approximately 1:1.

## A.3    CLASSIFIER FREE GUIDANCE

We apply classifier-free guidance for both generation and editing. For generation, we nullify the text prompt with a probability of 0.1 during training. For editing, we follow Step1X-Edit (Liu et al., 2025) to nullify the text prompt with a probability of 0.1 during training but do not nullify the reference image. All tasks use a guidance scale of 4.5.

## A.4    ADVERSARIAL TRAINING

In the second stage, we enable adversarial training (Karras et al., 2019) after 200k steps, introducing a discriminator to improve the reconstruction fidelity. The training loss is formulated as follows:

$$\mathcal{L}_{\text{stage2}} = \mathcal{L}_{\text{MSE}} + \mathcal{L}_{\text{LPIPS}} + \lambda \mathcal{L}_{\text{Distill}} + \lambda_G \mathcal{L}_{\text{GAN}}$$

where $\lambda_G$ is the weight for the discriminator loss $\mathcal{L}_{GAN}$, which is implemented as a binary cross-entropy (BCE) loss. We follow (Kim et al., 2025) to set the $\lambda_G$ to 0.1.

# B    ADDITIONAL MAIN RESULTS

## B.1    IMAGE RECONSTRUCTION

Both the 1B and 3B models employ InternViT-300M as the vision encoder and DC-AE as the pixel decoder, differing only in their intermediate MLP layers. In the 1B model, a two-layer MLP projects features from 1024 to 896 dimensions, followed by a three-layer MLP that maps them to 32 dimensions. The 3B model, in contrast, maps features from 1024 to 1536 dimensions before the final projection to 32. Therefore, the visual encoder and decoder parameters in the 1B and 3B models are very close, at 472M and 484M, respectively. The UniLIP in 3B model also achieves strong reconstruction performance, with an rFID of 0.35, a PSNR of 25.02, and an SSIM of 0.790.

## B.2    IMAGE GENERATION AND EDITING

We list full results of GenEval, WISE, and ImgEdit in Table 18, Table 19, Table 20.

## B.3    INFERENCE SPEED

As shown in Table 9, UniLIP is significantly faster at image generation than unified models with similar parameter counts, with all latencies measured on a single A100 GPU using the same text

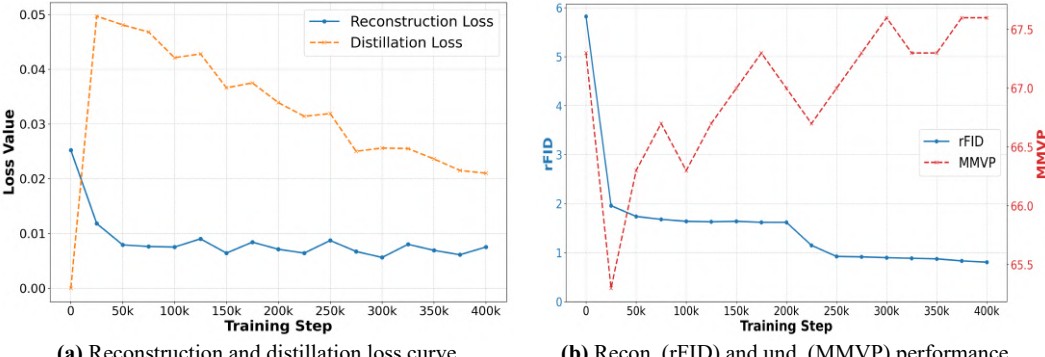

(a) Reconstruction and distillation loss curve.    (b) Recon. (rFID) and und. (MMVP) performance.

Figure 7: **Loss and performance curve in reconstruction training.**

| Model | Type | Resolution | Steps | Latency |
|---|---|---|---|---|
| Janus-Pro-1B | AR | 384 | 576 | 8.5 |
| Harmon-1.5B | MAR | 448 | 64 | 46.1 |
| BLIP3-o-4B | Diff. | 448 | 30+50 | 5.1 |
| UniLIP-1B | Diff. | 448 | 20 | 1.4 |
| UniLIP-3B | Diff. | 448 | 20 | 1.7 |

| Stage 1 | 2 | 3 | GenEval | WISE | ImgEdit |
|---|---|---|---|---|---|
| ✗ | ✓ | ✓ | 0.87 | 0.53 | 3.77 |
| ✓ | ✗ | ✓ | 0.84 | 0.49 | 3.58 |
| ✓ | ✓ | ✗ | 0.69 | 0.48 | 3.75 |
| ✓ | ✓ | ✓ | 0.88 | 0.56 | 3.81 |

Table 9: **Inference latency (s) comparison.**    Table 10: **Ablation of training stages.**

prompt. Compared to autoregressive methods, UniLIP requires far fewer iterations (steps). Furthermore, unlike BLIP3-o's complex, two-stage diffusion process, UniLIP generates images directly from its features via a lightweight pixel decoder, yielding a substantial speed advantage.

### B.4    REASONING EDITING

To comprehensively assess UniLIP's editing capabilities, we evaluated it on the RISE benchmark. Results in Table 12 show that our 3B model outperforms significantly larger models like Step1X-Edit (7B+12B) and Bagel (7B+7B), confirming UniLIP's superiority in reasoning-based editing.

### B.5    LINEAR AND ATTENTIVE PROBE

We conduct both linear probe and attentive probe evaluations on ImageNet. The results in Table 13 show that our UniLIP achieves classification performance comparable to the original InternViT. This validates the effectiveness of our reconstruction training and self-distillation loss.

## C    MORE ABLATIONS

### C.1    RECONSTRUCTION TRAINING CURVE

Figure 7 shows the stage two (224x224) training curves. Initially, the distillation loss rises sharply, suggesting interference with original understanding capabilities. After that, this loss consistently drops, while the understanding performance steadily improves to match the start level. Meanwhile, reconstruction loss remains stable and rFID steadily decreases (with a sharp drop at 200k steps due to adversarial training). We attribute this to the low information capacity required for expressing pixel details. For instance, a standard dimension of VAE is only 16, significantly smaller than CLIP. This allows strong reconstruction by making only slight adjustments to the original CLIP features (as evidenced by the distillation loss of 0.02). As a result, UniLIP effectively balances the dual objectives of understanding and reconstruction.

### C.2    GENERATION AND EDITING TRAINING STAGE

The training stage ablation in Table 10 confirms that all stages are essential. The third instruction-tuning stage significantly improves GenEval scores due to its dataset's similar prompt style. The

| Type | Weight | Reconstruction | | | Understanding | | | | |
|---|---|---|---|---|---|---|---|---|---|
| | | rFID↓ | PSNR↑ | SSIM↑ | MME-P↑ | MMBench↑ | MMVP↑ | AI2D↑ | TextVQA↑ |
| L1 | 1.0 | 0.32 | 24.55 | 0.786 | 1496 | 72.5 | 68.3 | 69.8 | 74.1 |
| Cosine | 1.0 | 0.35 | 24.82 | 0.798 | 1492 | 72.2 | 67.3 | 69.4 | 73.6 |
| MSE | 1.0 | 0.31 | 24.62 | 0.788 | 1499 | 72.6 | 68.7 | 70.7 | 74.7 |
| Cls | 1.0 | 0.36 | 25.09 | 0.805 | 820 | 53.6 | 54.7 | 58.4 | 53.0 |
| MSE | 0.1 | 0.28 | 25.03 | 0.801 | 1486 | 71.7 | 67.0 | 69.5 | 73.5 |
| MSE | 5.0 | 0.42 | 24.28 | 0.774 | 1511 | 72.8 | 68.3 | 70.6 | 74.3 |

Table 11: **Ablation of distillation loss types and weights.** "Cls" means classification loss.

| Model | Size | Temporal | Causal | Spatial | Logic | Overall |
|---|---|---|---|---|---|---|
| OmniGen (Xiao et al., 2025) | 3.8B | 1.2 | 1.0 | 0.0 | 1.2 | 0.8 |
| Step1X-Edit (Liu et al., 2025) | 7B + 12B | 0.0 | 2.2 | 2.0 | 3.5 | 1.9 |
| BAGEL (Deng et al., 2025) | 7B + 7B | 2.4 | 5.6 | 14.0 | 1.2 | 6.1 |
| **UniLIP-1B** | 1B+0.6B | 5.9 | 7.8 | 1.0 | 1.1 | 3.9 |
| **UniLIP-3B** | 2B + 1.6B | 9.4 | 15.5 | 5.0 | 2.4 | 8.1 |

Table 12: **Evaluation of image editing ability on RISE benchmark.**

| Model | Linear | Attentive |
|---|---|---|
| Frozen CLIP | 79.8 | 82.6 |
| **UniLIP** | 79.5 | 82.4 |

Table 13: **Linear and attentive probing results on ImageNet validation.**

second stage primarily affects WISE and ImgEdit, highlighting the need for both extensive pretraining and high-quality editing data. Finally, the initial alignment stage is also important, likely because it prevents the randomly initialized connector from disrupting pretrained capabilities.

### C.3 DISTILLATION LOSS

Table 11 ablates the distillation loss type and weight. At a fixed weight, MSE is most effective for preserving understanding, while classification loss is least effective because it provides only coarse, image-level supervision signals. Regarding weighting, low values are detrimental to understanding, yet high values offer no significant gains, likely due to overfitting that degrades generalization and reconstruction. Therefore, MSE with a weight of 1.0 achieves the optimal trade-off.

### C.4 UNDERSTANDING THE QUERY EMBEDDINGS

We employ learnable queries primarily to enable knowledge-augmented generation (e.g., for WISE benchmarks). In such tasks, prompts like "Draw Thanksgiving food" do not explicitly describe the visual details, so the model must rely on internal knowledge to deduce what to draw. To better illustrate the effect of query embeddings, we treat the output query embeddings as text embeddings and decode them into text. We observe that the decoded text can explicitly identify the target content. For instance, given the prompt "Thanksgiving food", the decoded text includes the word "turkey", which greatly simplifies generation for the DiT. In contrast, models lacking this mechanism generate incorrect foods. We provide detailed visual comparisons in Figure 8.

### C.5 QUERY LENGTH

We investigate the impact of the learnable query length, denoted as $L$, within our dual-condition architecture. As presented in Table 14, the performance on the WISE benchmark exhibits a positive correlation with query length, reaching a point of convergence around $L = 256$. We hypothesize that longer queries effectively mimic an increased inference budget for textual reasoning. This expanded capacity enables the Large Language Model (LLM) to more comprehensively activate its internal knowledge, thereby facilitating more accurate image generation.

| Query Length | Cultural | Time | Space | Biology | Physics | Chemistry. | Overall |
|---|---|---|---|---|---|---|---|
| 0 | 0.43 | 0.50 | 0.62 | 0.44 | 0.58 | 0.35 | 0.47 |
| 64 | 0.48 | 0.52 | 0.62 | 0.42 | 0.54 | 0.40 | 0.50 |
| 128 | 0.52 | 0.56 | 0.68 | 0.48 | 0.58 | 0.45 | 0.54 |
| 256 | 0.54 | **0.58** | **0.70** | **0.50** | **0.62** | **0.46** | **0.56** |
| 512 | **0.56** | 0.57 | 0.67 | 0.48 | 0.60 | 0.44 | **0.56** |

Table 14: Ablation study on the impact of **learnable query** length on the WISE benchmark.

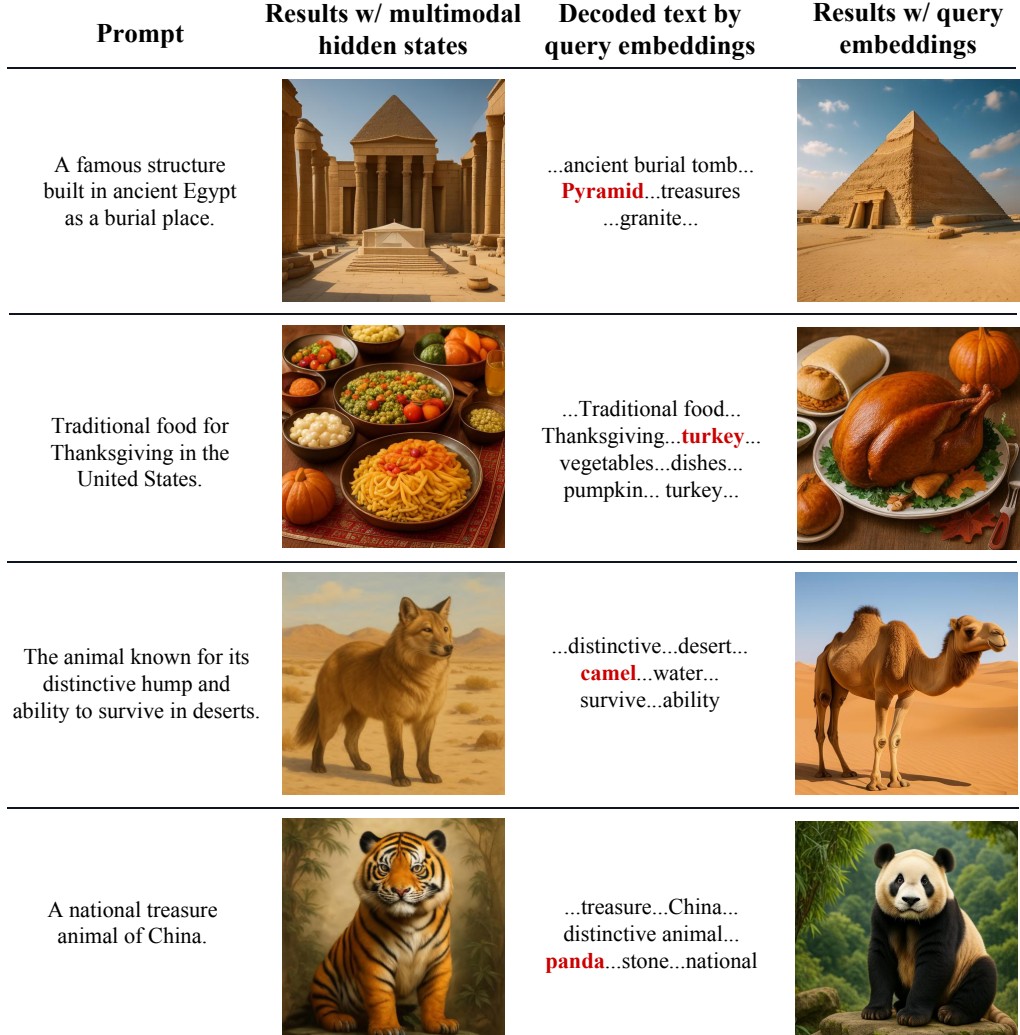

Figure 8: **Qualitative comparison between multimodal hidden states and query embeddings.** We decode query embeddings back into text (masking unreadable tokens with ...). The results show high semantic consistency with both the prompt and the generated image, especially the red-highlighted terms.

| Method | Add | Adj. | Ext. | Repl. | Rmv. | Bkg. | Style | Hyb. | Act. | Overall |
|---|---|---|---|---|---|---|---|---|---|---|
| Edit Only | 3.99 | 3.38 | 1.77 | 4.16 | 3.92 | 3.88 | 4.75 | 2.96 | 4.21 | 3.67 |
| **Joint Training** | **4.11** | **3.58** | **1.94** | **4.30** | **3.97** | **4.00** | **4.87** | **3.17** | **4.35** | **3.81** |

Table 15: **Ablation study on training strategies.** We compare a model trained solely on editing data against our joint training strategy. The joint approach leverages large-scale generation data to enhance editing performance across various sub-tasks.

### C.6 JOINT TRAINING

We unify image generation and editing to exploit task synergy, specifically enabling generation capabilities to facilitate editing tasks. Given the scarcity of high-quality editing data (1.5M samples) compared to generation data (36M), training solely on the former poses a significant risk of overfitting. By incorporating large-scale generation data, the model captures a more robust underlying image distribution, leading to more realistic and consistent edits. To validate this, we train a 1B-parameter variant using only the editing dataset. As demonstrated in Table 15, joint training significantly outperforms the "Edit Only" baseline across all metrics, confirming that generative knowledge is crucial for high-quality editing.

### C.7 COMPARISON WITH DUAL-ENCODER METHODS

To demonstrate the superiority of our unified encoder over the dual-encoder architecture, we conduct ablation studies based on JanusFlow (Ma et al., 2025b). We construct a unified variant, UniLIP-JanusFlow, by replacing the original dual encoders (SigLIP and SDXL-VAE) with our UniLIP-SigLIP. Specifically, we first apply our two-stage reconstruction training to the SigLIP encoder within JanusFlow to enable reconstruction (UniLIP-SigLIP). Subsequently, using JanusFlow's pre-trained weights, we fine-tune both the original dual-encoder baseline (JanusFlow) and our unified model (UniLIP-JanusFlow) on the ShareGPT-4o-Image Chen et al. (2025b) dataset for 10k steps.

We first evaluate the fundamental capabilities of the encoders as shown in Table 16. In terms of reconstruction, UniLIP-SigLIP achieves a lower rFID (0.35) than the original SDXL-VAE (0.38) while maintaining comparable PSNR and SSIM, proving it can serve as a high-quality visual generator. Crucially, this added capability does not compromise semantic understanding; UniLIP-SigLIP preserves or slightly improves performance on multimodal benchmarks (e.g., MME-P, MMVP) compared to the original SigLIP. This confirms that our approach effectively resolves the conflict between reconstruction and understanding tasks.

| Model | Reconstruction | | | Understanding | | | | |
|---|---|---|---|---|---|---|---|---|
| | rFID↓ | PSNR↑ | SSIM↑ | MME-P↑ | MMBench↑ | MMVP↑ | AI2D↑ | TextVQA↑ |
| JanusFlow (Ma et al., 2025b) | 0.38 | **26.79** | **0.827** | 1302.0 | 64.6 | 65.0 | 65.7 | 55.5 |
| **UniLIP-JanusFlow** | **0.35** | 26.04 | 0.803 | **1305.4** | **64.8** | **65.7** | **65.8** | **55.8** |

Table 16: **Reconstruction and understanding performance comparisons**.

Next, we compare the performance on generation and editing tasks (Table 17). To extend JanusFlow for the editing task, we follow previous work (Chen et al., 2025b) by concatenating SigLIP and VAE features, resulting in a total of 1152 image tokens. In contrast, UniLIP-JanusFlow utilizes a single unified feature sequence of only 576 tokens. Despite the significant reduction in token count, UniLIP-JanusFlow consistently outperforms the dual-encoder baseline (+2.0 on GenEval, +5.0 on WISE, and +0.14 on ImgEdit). These results demonstrate that the unified representation eliminates the feature misalignment inherent in dual-encoder systems and provides a more efficient and effective foundation for generative tasks.

## D QUALITATIVE COMPARISONS

### D.1 IMAGE RECONSTRUCTION

Figure 9 illustrates UniLIP's superior reconstruction against other methods. Pairing a simple pixel decoder with frozen CLIP yields blurry outputs (third column). While Emu2 (Sun et al., 2024b) gen-

| Model | GenEval | | | WISE | | | ImgEdit | | |
|---|---|---|---|---|---|---|---|---|---|
| | Sgl. Obj. | Two Obj. | **Overall** | Time | Space | **Overall** | Add | Repl. | **Overall** |
| JanusFlow (Ma et al., 2025b) | 0.99 | 0.82 | 0.73 | 0.35 | 0.46 | 0.31 | 3.21 | 2.92 | 2.95 |
| **UniLIP-JanusFlow** | **1.00** | **0.85** | **0.75** | **0.40** | **0.49** | **0.36** | **3.38** | **3.15** | **3.09** |

Table 17: **Generation and Editing Performance.** Models are fine-tuned on ShareGPT-4o-Image. We report representative sub-metrics (e.g., Single/Two Object for GenEval, Time/Space for WISE, Add/Replace for ImgEdit) alongside the overall scores. UniLIP-JanusFlow consistently outperforms the dual-encoder baseline across both fine-grained and aggregate metrics.

erates clearer images by conditioning a diffusion model on CLIP features, it sacrifices consistency due to CLIP's inherent information loss, evidenced by altered clock digits. TokenFlow (Qu et al., 2024) also exhibits blurring on faces and text because of information loss from feature quantization. Conversely, UniLIP employs a carefully designed training strategy for CLIP and preserves continuous features, enabling it to deliver reconstruction quality matching DC-AE (Chen et al., 2024b).

### D.2    IMAGE GENERATION

Figure 10 shows the qualitative comparisons of image generation. UniLIP accurately renders positional relationships and clear faces (first row), unlike Harmon (Wu et al., 2025c) (reversed positions) or other models (distortions/obscured faces). Our semantic reasoning excels in the fourth row, uniquely generating "the fastest land animal." Furthermore, UniLIP demonstrates superior aesthetic sensibility, with its final row output more faithfully capturing Monet's style than other methods.

### D.3    IMAGE EDITING

Figure 11 highlights UniLIP's superior editing, excelling in instruction adherence and consistency. For instance, UniLIP uniquely extracts the motorcycle correctly in the third row, where other models revert or erase the object. It also generates more natural content: in the final row, UniLIP seamlessly inpaints the erased car area based on background context, avoiding other models' prominent blurring artifacts and yielding a much more natural result.

### D.4    ABLATION OF REFERENCE IMAGE ENCODER

Figure 12 compares the editing results of CLIP and UniLIP. While CLIP successfully follows the editing instruction, it struggles to preserve the subject's identity, texture, and background. In contrast, UniLIP leverages reconstruction training to overcome this limitation, achieving greater fidelity.

## E    LIMITATION AND FUTURE WORK

Currently, the scale of UniLIP is limited to 3B, which is significantly smaller than models like Step1X-Edit (19B) and Qwen-Image (28B). Besides, UniLIP can only support $448 \times 448$ resolution. These limitations result in subpar performance on text rendering and editing, a common flaw in many existing approaches (Xie et al., 2025; Wu et al., 2025b; Han et al., 2025). In the future, we will further scale up the model size and resolution to enhance text rendering-related tasks.

## F    THE USE OF LARGE LANGUAGE MODELS

During the preparation of this manuscript, we utilize Large Language Models (LLMs) solely for the purpose of language polishing, grammar correction, and improving readability. All core research aspects, including ideation, data analysis, and the formulation of conclusions, are conducted entirely by the authors. The authors take full responsibility for the final content of the manuscript.

| Original | UniLIP | Frozen CLIP | Emu2 | TokenFlow | DC-AE |
|----------|--------|-------------|------|-----------|-------|

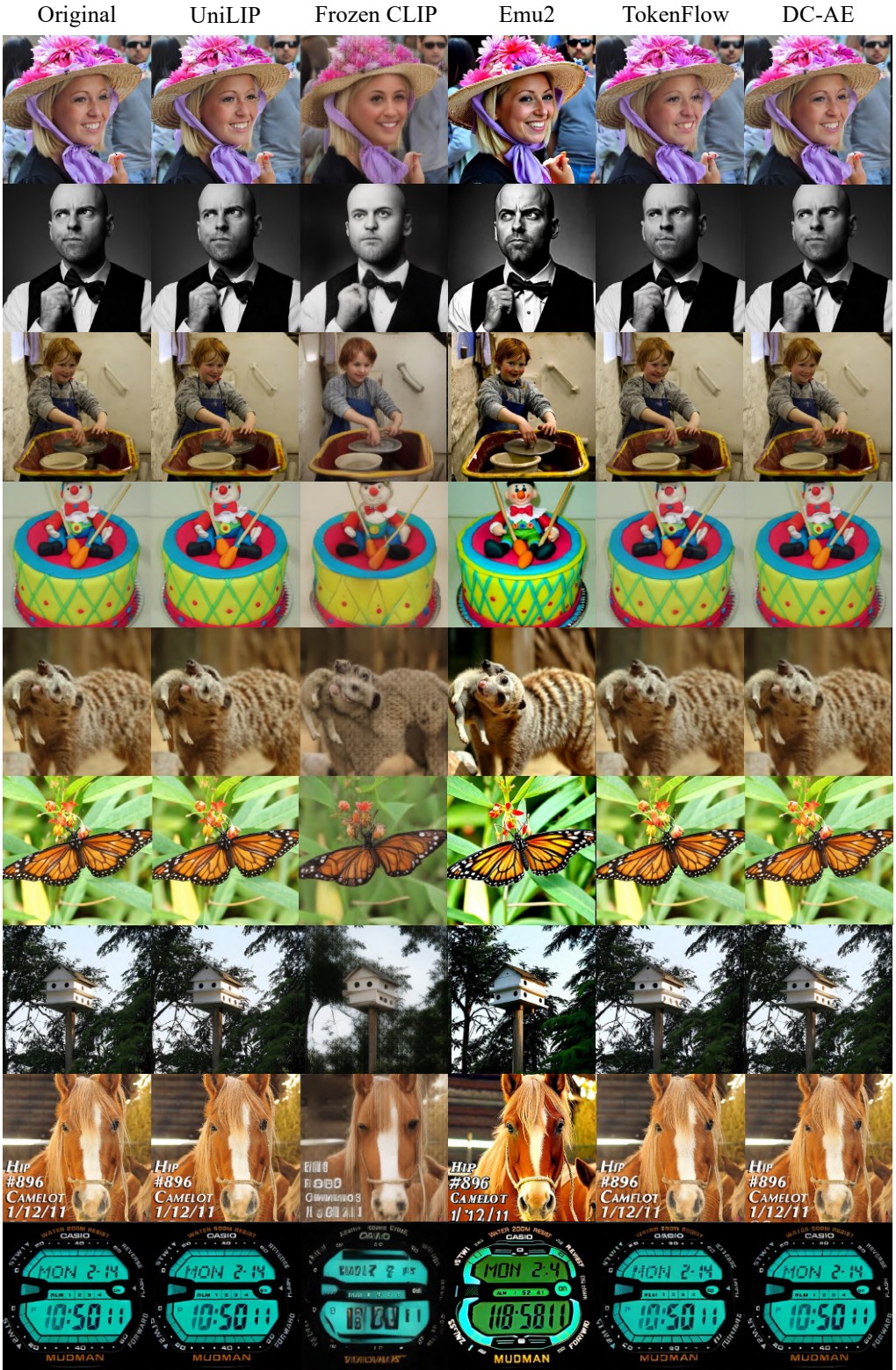

Figure 9: **Qualitative comparison of image reconstruction.**

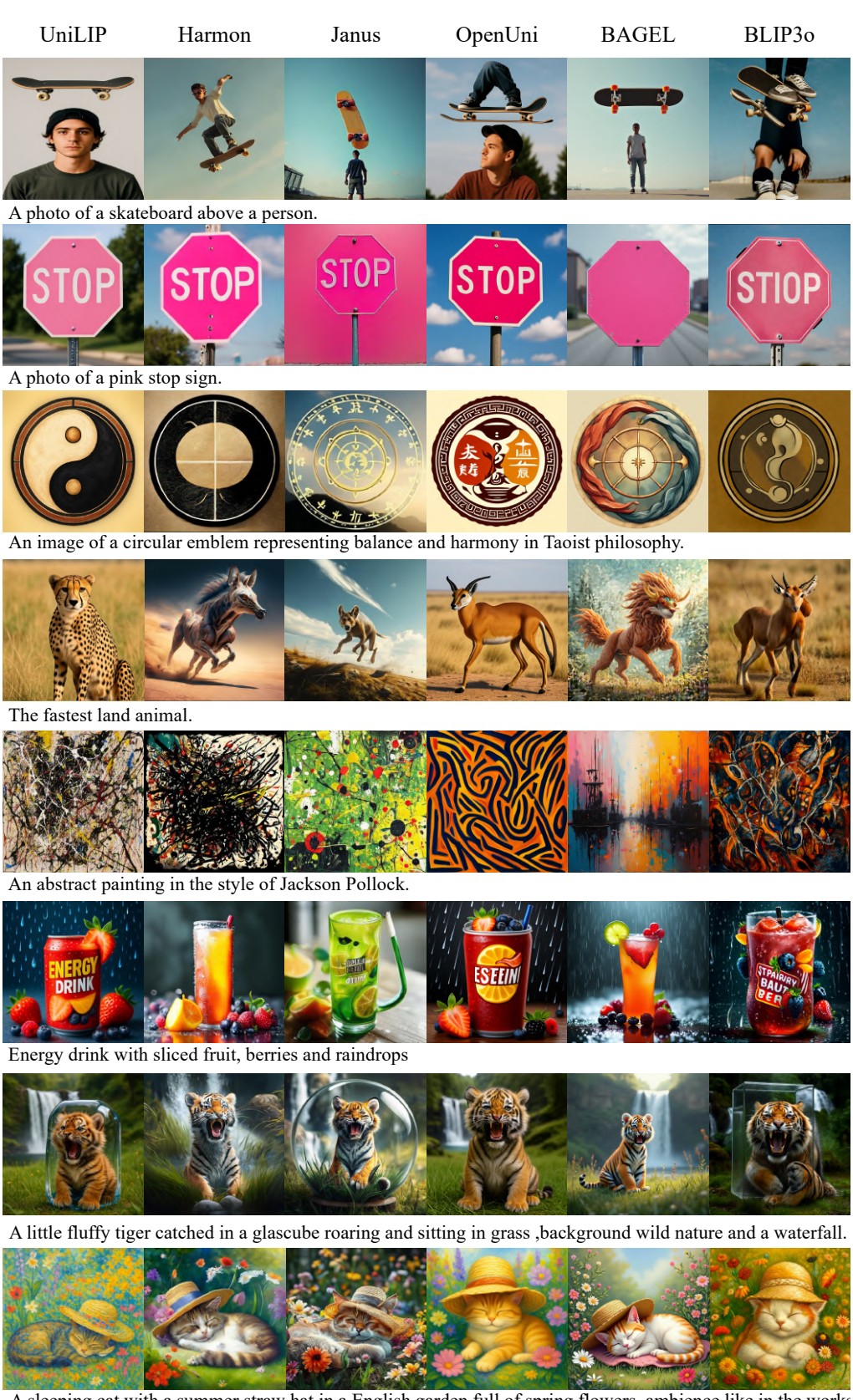

Figure 10: **Qualitative comparison of text-to-image generation.**

Reference    UniLIP    UniWorld-V1    Step1X-Edit    BAGEL    OmniGen2

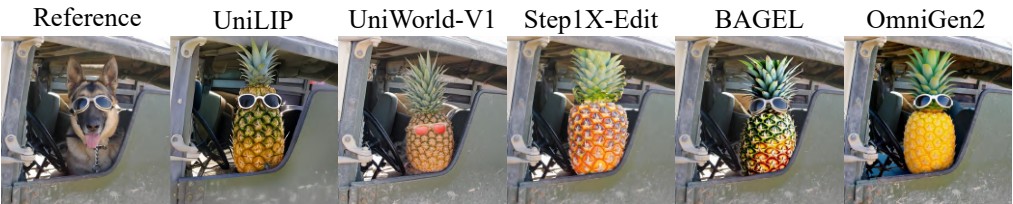

Replace the dog in the image with a large pineapple while keeping it seated and wearing the same goggles.

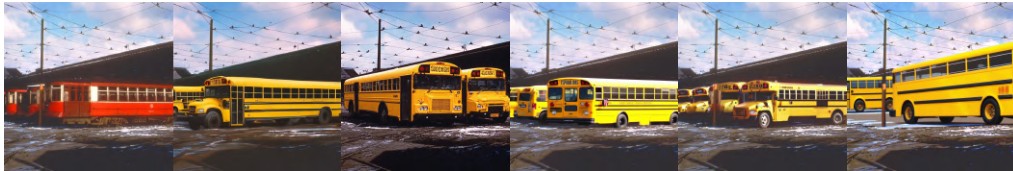

Replace the red trams in the image with yellow school buses.

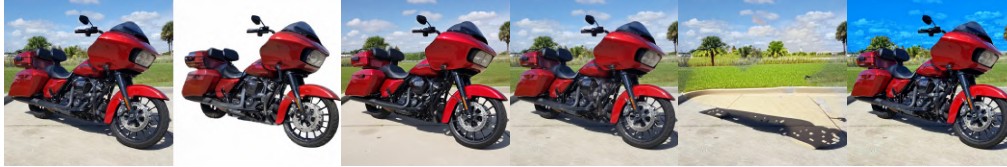

Extract the red motorcycle parked on the concrete surface in the image.

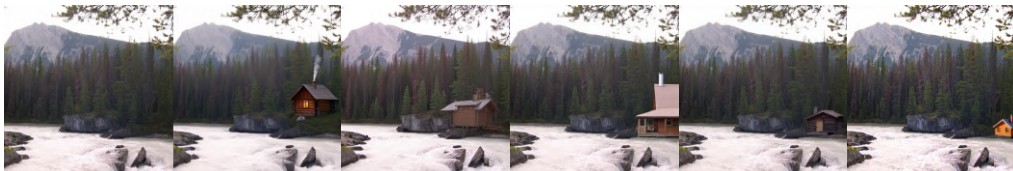

Add a small wooden cabin with a chimney near the edge of the forest on the right side of the image.

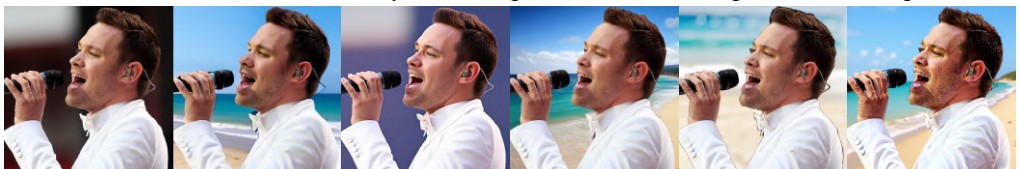

Change the concert stage background to a beach setting.

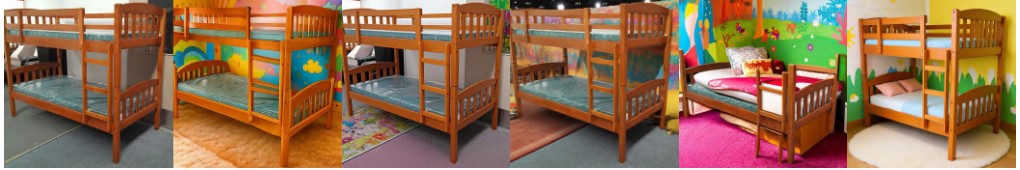

Change the indoor showroom environment in the picture to a cozy children's bedroom with colorful wall murals and plush carpets.

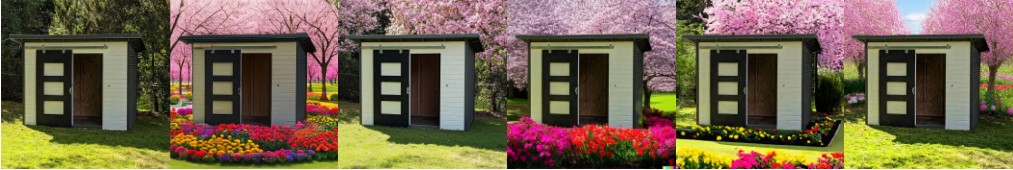

Add a flower garden surrounding the shed and change the trees in the background to blossom trees.

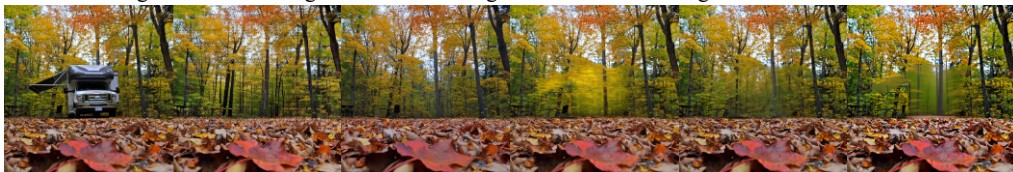

Remove the camper van in the image and make the surrounding forest more seamless.

Figure 11: **Qualitative comparison of image editing.**

| Model | Single Obj. | Two Obj. | Counting | Colors | Position | Color Attri. | Overall |
|---|---|---|---|---|---|---|---|
| *Gen. Only* | | | | | | | |
| DALLE-2 (Ramesh et al., 2022) | 0.94 | 0.66 | 0.49 | 0.77 | 0.10 | 0.19 | 0.52 |
| SDv2.1 (Rombach et al., 2022) | 0.98 | 0.51 | 0.44 | 0.85 | 0.07 | 0.17 | 0.50 |
| DALLE-3 (Betker et al., 2023) | 0.96 | 0.87 | 0.47 | 0.83 | 0.43 | 0.45 | 0.67 |
| SDXL (Podell et al., 2023) | 0.98 | 0.74 | 0.39 | 0.85 | 0.15 | 0.23 | 0.55 |
| FLUX.1-dev (Labs, 2023) | 0.98 | 0.93 | 0.75 | 0.93 | 0.68 | 0.65 | 0.82 |
| PixArt-$\alpha$ (Chen et al., 2024a) | 0.98 | 0.50 | 0.44 | 0.80 | 0.08 | 0.07 | 0.48 |
| Emu3-Gen (Wang et al., 2024c) | 0.98 | 0.71 | 0.34 | 0.81 | 0.17 | 0.21 | 0.54 |
| SD3-Medium (Esser et al., 2024) | 0.99 | 0.94 | 0.72 | 0.89 | 0.33 | 0.60 | 0.74 |
| Sana-1.6B (Xie et al., 2024a) | 0.99 | 0.77 | 0.62 | 0.82 | 0.21 | 0.47 | |
| *Und. and Gen.* | | | | | | | |
| LWM-7B (Liu et al., 2024b) | 0.93 | 0.41 | 0.46 | 0.79 | 0.09 | 0.15 | 0.47 |
| SEED-X-13B (Ge et al., 2024) | 0.97 | 0.58 | 0.26 | 0.80 | 0.19 | 0.14 | 0.49 |
| TokenFlow-XL (Qu et al., 2024) | 0.95 | 0.60 | 0.41 | 0.81 | 0.16 | 0.24 | 0.55 |
| ILLUME (Wang et al., 2024a) | 0.99 | 0.86 | 0.45 | 0.71 | 0.39 | 0.28 | 0.61 |
| ILLUME+ (Huang et al., 2025) | 0.99 | 0.88 | 0.62 | 0.84 | 0.42 | 0.53 | 0.72 |
| Emu3-Gen-8B (Wang et al., 2024c) | 0.99 | 0.81 | 0.42 | 0.80 | 0.49 | 0.45 | 0.66 |
| Janus-Pro-7B (Chen et al., 2025c) | 0.99 | 0.89 | 0.59 | 0.90 | 0.79 | 0.66 | 0.80 |
| MetaQuery-B (Pan et al., 2025) | - | - | - | - | - | - | 0.74 |
| MetaQuery-XL (Pan et al., 2025) | - | - | - | - | - | - | 0.80 |
| Harmon-1.5B (Wu et al., 2025c) | 0.99 | 0.86 | 0.66 | 0.85 | 0.74 | 0.48 | 0.76 |
| BLIP3-o-4B (Chen et al., 2025a) | - | - | - | - | - | - | 0.81 |
| BLIP3-o-8B (Chen et al., 2025a) | - | - | - | - | - | - | 0.84 |
| BAGEL (Deng et al., 2025) | 0.99 | 0.94 | 0.81 | 0.88 | 0.64 | 0.63 | 0.82 |
| OpenUni-B (Wu et al., 2025b) | 0.99 | 0.91 | 0.74 | 0.90 | 0.77 | 0.73 | 0.84 |
| OpenUni-L (Wu et al., 2025b) | 0.99 | 0.91 | 0.77 | 0.90 | 0.75 | 0.76 | 0.85 |
| Tar (Han et al., 2025) | 0.98 | 0.92 | 0.83 | 0.85 | 0.80 | 0.65 | 0.84 |
| **UniLIP-1B** | 0.99 | 0.92 | 0.83 | 0.91 | 0.83 | 0.80 | 0.88 |
| **UniLIP-3B** | 1 | **0.95** | **0.84** | **0.91** | **0.86** | **0.84** | **0.90** |

Table 18: **Evaluation of text-to-image generation ability on GenEval benchmark.**

| Model | Cultural | Time | Space | Biology | Physics | Chemistry. | Overall |
|---|---|---|---|---|---|---|---|
| *Gen. Only* | | | | | | | |
| SDv1.5 (Rombach et al., 2022) | 0.34 | 0.35 | 0.32 | 0.28 | 0.29 | 0.21 | 0.32 |
| SDXL (Podell et al., 2023) | 0.43 | 0.48 | 0.47 | 0.44 | 0.45 | 0.27 | 0.43 |
| FLUX.1-dev (Labs, 2023) | 0.48 | 0.58 | 0.62 | 0.42 | 0.51 | 0.35 | 0.50 |
| PixArt-$\alpha$ (Chen et al., 2024a) | 0.45 | 0.50 | 0.48 | 0.49 | 0.56 | 0.34 | 0.47 |
| SD3.5-large (Esser et al., 2024) | 0.44 | 0.50 | 0.58 | 0.44 | 0.52 | 0.31 | 0.46 |
| Playground-v2.5 (Li et al., 2024c) | 0.49 | 0.58 | 0.55 | 0.43 | 0.48 | 0.33 | 0.49 |
| *Und. and Gen.* | | | | | | | |
| VILA-U-7B (Wu et al., 2024b) | 0.26 | 0.33 | 0.37 | 0.35 | 0.39 | 0.23 | 0.31 |
| Janus-Pro-7B (Chen et al., 2025c) | 0.30 | 0.37 | 0.49 | 0.36 | 0.42 | 0.26 | 0.35 |
| Emu3-Gen-8B (Wang et al., 2024c) | 0.34 | 0.45 | 0.48 | 0.41 | 0.45 | 0.27 | 0.39 |
| MetaQuery-B (Pan et al., 2025) | 0.44 | 0.49 | 0.58 | 0.41 | 0.49 | 0.34 | 0.46 |
| MetaQuery-XL (Pan et al., 2025) | 0.56 | 0.55 | 0.62 | 0.49 | 0.63 | 0.41 | 0.55 |
| Harmon-1.5B (Wu et al., 2025c) | 0.38 | 0.48 | 0.52 | 0.37 | 0.44 | 0.29 | 0.41 |
| BLIP3-o-4B (Chen et al., 2025a) | - | - | - | - | - | - | 0.50 |
| BLIP3-o-4B (Chen et al., 2025a) | - | - | - | - | - | - | 0.62 |
| BAGEL (Deng et al., 2025) | 0.44 | 0.55 | 0.68 | 0.44 | 0.60 | 0.39 | 0.52 |
| OpenUni-B (Wu et al., 2025b) | 0.37 | 0.45 | 0.58 | 0.39 | 0.50 | 0.30 | 0.43 |
| OpenUni-L (Wu et al., 2025b) | 0.51 | 0.49 | 0.64 | 0.48 | 0.63 | 0.35 | 0.52 |
| **UniLIP-1B** | 0.54 | 0.58 | 0.70 | 0.50 | 0.62 | 0.46 | 0.56 |
| **UniLIP-3B** | **0.66** | **0.60** | **0.70** | **0.60** | **0.68** | **0.43** | **0.63** |

Table 19: **Comparison of world knowledge reasoning on WISE.**

| Model | Add | Adj. | Ext. | Repl. | Rmv. | Bkg. | Style | Hyb. | Act. | Overall |
|---|---|---|---|---|---|---|---|---|---|---|
| GPT-4o (OpenAI, 2025) | 4.61 | 4.33 | 2.9 | 4.35 | 3.66 | 4.57 | 4.93 | 3.96 | 4.89 | 4.20 |
| MagicBrush (Zhang et al., 2023) | 2.84 | 1.58 | 1.51 | 1.97 | 1.58 | 1.75 | 2.38 | 1.62 | 1.22 | 1.90 |
| Instruct-P2P (Brooks et al., 2023) | 2.45 | 1.83 | 1.44 | 2.01 | 1.50 | 1.44 | 3.55 | 1.20 | 1.46 | 1.88 |
| AnyEdit (Yu et al., 2025) | 3.18 | 2.95 | 1.88 | 2.47 | 2.23 | 2.24 | 2.85 | 1.56 | 2.65 | 2.45 |
| UltraEdit (Zhao et al., 2024) | 3.44 | 2.81 | 2.13 | 2.96 | 1.45 | 2.83 | 3.76 | 1.91 | 2.98 | 2.70 |
| OmniGen (Xiao et al., 2025) | 3.47 | 3.04 | 1.71 | 2.94 | 2.43 | 3.21 | 4.19 | 2.24 | 3.38 | 2.96 |
| Step1X-Edit (Liu et al., 2025) | 3.88 | 3.14 | 1.76 | 3.40 | 2.41 | 3.16 | 4.63 | 2.64 | 2.52 | 3.06 |
| ICEdit (Zhang et al., 2025) | 3.58 | 3.39 | 1.73 | 3.15 | 2.93 | 3.08 | 3.84 | 2.04 | 3.68 | 3.05 |
| BAGEL (Deng et al., 2025) | 3.56 | 3.31 | 1.70 | 3.30 | 2.62 | 3.24 | 4.49 | 2.38 | 4.17 | 3.20 |
| UniWorld-V1 (Lin et al., 2025a) | 3.82 | 3.64 | 2.27 | 3.47 | 3.24 | 2.99 | 4.21 | 2.96 | 2.74 | 3.26 |
| Janus-4o (Chen et al., 2025b) | 3.60 | 3.25 | 2.28 | 3.27 | 2.28 | 3.32 | 4.47 | 2.74 | 4.13 | 3.26 |
| OmniGen2 (Wu et al., 2025a) | 3.57 | 3.06 | 1.77 | 3.74 | 3.20 | 3.57 | 4.81 | 2.52 | **4.68** | 3.44 |
| **UniLIP-1B** | 4.11 | 3.58 | 1.94 | 4.30 | 3.97 | 4.00 | **4.87** | **3.17** | 4.35 | 3.81 |
| **UniLIP-3B** | **4.29** | **3.90** | **2.22** | **4.44** | **4.10** | **4.14** | 4.80 | 3.04 | 4.55 | **3.94** |

Table 20: **Evaluation of image editing ability on ImgEdit benchmark.**

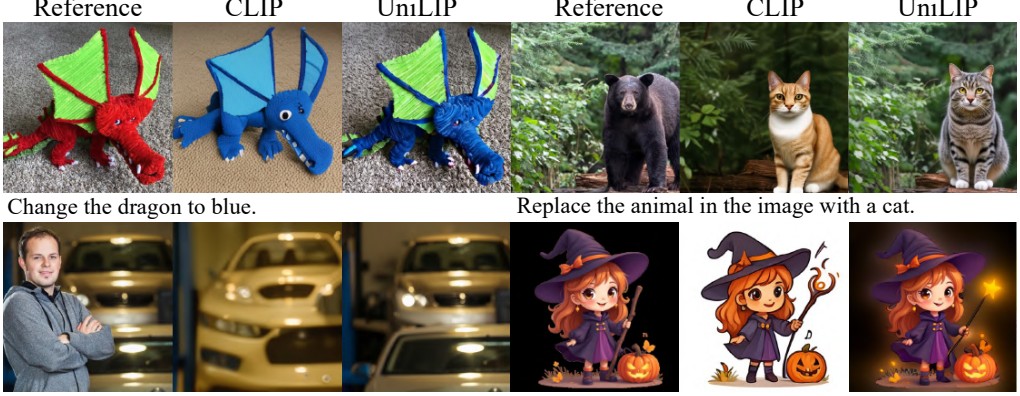

Figure 12: **Qualitative comparison between CLIP and UniLIP as reference image encoder.**

