# OpenReview forum: "UniLiP: Adapting CLIP for Unified Multimodal Understanding, Generation and Editing"
_ICLR.cc/2026/Conference — ICLR 2026 Poster_

### Official Review · Reviewer_E7dY · 2025-10-22

**Soundness:** 3
**Presentation:** 3
**Contribution:** 3
**Rating:** 6
**Confidence:** 3

**Summary:**

This paper introduces UniLIP, a unified framework that extends CLIP to support multimodal tasks, including understanding, generation, and editing. UniLIP incorporates a two-phase training methodology featuring self-distillation, which enhances CLIP's capabilities for high-fidelity reconstruction without compromising its baseline comprehension performance. To improve reasoning and ensure consistency in generation and editing tasks, the authors have innovatively implemented a dual-condition architecture within the MetaQuery framework. Despite its relatively smaller size with 1B and 3B parameters, UniLIP demonstrates superior performance over larger models such as BAGEL (7B) and Uniworld-V1 (12B), achieving benchmark scores of 0.90 on GenEval, 0.63 on WISE, and 3.94 on ImgEdit.

**Strengths:**

1. This paper effectively addresses the limitations in CLIP's reconstruction capabilities through a two-stage training process, transforming it into a unified visual encoder.
2. This proposed dual-condition architecture utilizes the multimodal hidden states and query embeddings as the input, which can effectively avoid information loss while leveraging the reasoning capabilities of LLMs through query embeddings.
3. This paper conducts extensive experiments on understanding, generation, and image editing. With only 1B and 3B parameters, UniLIP outperforms larger unified models such as BAGEL (7B) and Uniworld-V1 (12B), thereby demonstrating the efficacy of the proposed method.

**Weaknesses:**

1. Given that the dual-condition architecture is based on MetaQuery, the novelty appears limited.
2. In Tab.6, the performance improvement in understanding appears to be primarily due to self-distillation. Additionally, "Lr Decay" seems to have a more significant impact compared to "Two Stage."
3. The ablation studies in Table 6 are not thorough. For instance, they fail to convincingly demonstrate the effectiveness of each stage in the two-stage reconstruction training.

**Questions:**

1. After undergoing two-stage reconstruction training, how does UniLIP perform on downstream tasks such as linear probe or attentive probe within the CLIP framework?
2. Has the study explored any self-distillation losses other than MSE loss?

---

> ### Author Response · Authors · 2025-11-21
> **Rebuttal by Authors**
>
> Dear Reviewer E7dY,
>
> We sincerely appreciate your recognition of our method's effectiveness, as well as our performance in generation and editing. We address your questions below.
>
> > [W1] The novelty compared to MetaQuery
>
> While we adopt the MetaQuery framework, our approach introduces several key innovations:
>
> 1. We propose a **two-stage reconstruction framework with self-distillation** that transforms the MLLM vision encoder into a unified encoder, preserving understanding capabilities while achieving high-quality reconstruction.
> 2. Unlike MetaQuery, which utilizes VAE features as generation targets, we employ **semantically rich visual representations**. This results in superior alignment with text prompts.
> 3. Our **dual-condition architecture** is the first to effectively extend this framework to editing, demonstrating superior quantitative performance rather than only qualitative visualizations.
>
>
>
> > [W2] The impact of reconstruction training strategies
>
> While self-distillation is pivotal for preserving understanding, all components are essential; removing any single strategy degrades final performance. We appreciate your insight regarding Learning Rate (LR) decay and have emphasized its importance in the revised Method section.
>
>
>
> > [W3] The effectiveness of two-stage training
>
> Thanks for noting this important design.
> As shown in Table 6, two-stage training contributes to preserving understanding capabilities, albeit to a lesser extent than self-distillation. Beyond this, the two-stage strategy offers two key advantages:
>
> **1. Improved Stability and Faster Convergence.** Stage 1 pre-aligns the pixel decoder with the frozen CLIP, minimizing feature shifts when the encoder is unfrozen in Stage 2. Without this preliminary stage, the initial misalignment between CLIP and the decoder, and the randomly initialized projection layer will result in gradient instability during early training. In the revision, we draw **Figure 6** to compare the loss and performance of single-stage vs. two-stage training. The single-stage approach exhibits a large spike (**0.0939 vs. 0.0497**). Consequently, the convergence is significantly slower (**3× slower for distillation loss and 4× slower for performance recovery**).
>
> **2. Superior Training Efficiency.** Stage 2 is computationally expensive (approximately 3× slower than Stage 1) due to the unfrozen CLIP and self-distillation overhead. In Table 6, while the total iterations are the same, the single-stage method (which essentially runs as "Stage 2" the entire time) requires more training time. As detailed below, the two-stage strategy reduces total training time from 120h to 86h (on 4 A100 GPUs).
>
> | Steps/Time   | Stage 1  | Stage 2   | Total Time |
> | ------------ | -------- | --------- | ---------- |
> | Single Stage | /        | 700k/120h | 120h       |
> | Two Stage    | 300k/18h | 400k/68h  | 86h        |
>
> We also include the discussion above in Section 4.3 in revision.
>
> > [Q1] Downstream performance for CLIP
>
> We conduct both linear probe and attentive probe evaluations on ImageNet. The results below show that our UniLIP achieves classification performance comparable to the original InternViT. This validates the effectiveness of our reconstruction training and self-distillation loss in preserving semantic features.
>
> |                         | Linear | Attentive |
> | ----------------------- | ------ | --------- |
> | Frozen CLIP (InternViT) | 79.8   | 82.6      |
> | UniLIP                  | 79.5   | 82.4      |
>
>
>
> > [Q2] Self-distillation loss other than MSE loss
>
> We have included an evaluation of alternative self-distillation losses, including KL divergence and cosine similarity, in **Table 11 of the Appendix**. We also paste it here for your convenience. Empirically, MSE loss is the most effective for preserving understanding capabilities. This is likely because MSE imposes **stricter constraints**, whereas cosine similarity only aligns feature directions.
>
> | **Type**   | **rFID ↓** | **PSNR ↑** | **SSIM ↑** | **MME-P ↑** | **MMBench ↑** | **MMVP ↑** | **AI2D ↑** | **TextVQA ↑** |
> | ---------- | ---------- | ---------- | ---------- | ----------- | ------------- | ---------- | ---------- | ------------- |
> | **L1**     | 0.32       | 24.55      | 0.786      | 1496        | 72.5          | 68.3       | 69.8       | 74.1          |
> | **Cosine** | 0.35       | **24.82**  | **0.798**  | 1492        | 72.2          | 67.3       | 69.4       | 73.6          |
> | **MSE**    | **0.31**   | 24.62      | 0.788      | **1499**    | **72.6**      | 68.7       | **70.7**   | **74.7**      |
>
> ***
>
> Thanks for your professional comments. We hope our response addresses your questions. Feel free to let us know if you have any further questions or concerns :-).

---

> > ### Comment · Reviewer_E7dY · 2025-11-27
> >
> > Thanks for your response, which has addressed my concerns. Taking into account the comments from other reviewers as well as your reply, I'm inclined to maintain my current positive score and have adjusted my confidence.

---

> > > ### Author Response · Authors · 2025-11-28
> > >
> > > We sincerely thank you for your positive feedback and for acknowledging that your concerns have been addressed. As for the dual-encoder issue mentioned by Reviewer eMyq, we have provided more rigorous experimental comparisons. We believe these additional results further strengthen the effectiveness of our method.

---

### Official Review · Reviewer_6SkM · 2025-10-29

**Soundness:** 3
**Presentation:** 3
**Contribution:** 3
**Rating:** 6
**Confidence:** 3

**Summary:**

This paper proposes UniLip, a novel two-stage training scheme for CLIP to capture both semantic information and pixel details, enabling unified multimodal understanding and generation. The scheme finetunes the CLIP model to optimize reconstruction quality while preserving the features in the original CLIP. Moreover, this paper proposes a dual-condition architecture for editing tasks, which supplements hidden states of MLLMs as conditions to provide pixel details.

**Strengths:**

This paper presents a training approach for CLIP that effectively balances semantic understanding and pixel-level detail, achieving SOTA performance in various multimodal tasks.
The presentation is clear and well-structured, with comprehensive experiments and ablation studies demonstrating the effectiveness of the proposed method.

**Weaknesses:**

The description of related works for unified multimodal models could be more detailed to better position the contributions of this work.
The method requires finetuning for the pixel decoder in both stages; the reason for this design choice is not well explained.

**Questions:**

1. What is the benefit of unified CLIP for both generation and editing tasks, compared to training two separate models for these two tasks respectively to achieve similar performance?
2. Comparison with BLIP3-o: What is the essential reason for BLIP3-o applying a complex pipeline (described in lines 147-151)?
3. Why the stage 1, which solely trains the pixel decoder is necessary? Can we directly train both the CLIP and decoder module and skip stage 1? (The benefit of the two-stage is not significant in Table 6, maybe we can just increase the training time in one stage.)
4. The generation and editing require a different training process; why not describe the training in the method section?
5. How about other CLIP methods (e.g., BLIP3-o mentioned in the paper) perform on the reconstruction tasks?

---

> ### Author Response · Authors · 2025-11-21
> **Rebuttal by Authors**
>
> Dear Reviewer 6SkM,
>
> Many thanks for your valuable comments and questions, which help us a lot to improve our paper. We address your questions as follows.
>
> > [W1] More detailed related work for unified multimodal models
>
> Thanks for the valuable suggestions. In the revision, we have divided the related work for unified models into two parts: **Unified Tokenizer** and **Unified Architecture**, which better illustrate the advantages of UniLIP compared to previous works.
>
> > [Q1] The benefit of unified CLIP for generation and editing tasks.
>
> We unify these two tasks to **leverage task synergy**, specifically allowing generation to enhance editing. Editing is also a generative process: for instance, replacing a cat with a dog requires generating a new dog. Since high-quality editing data is scarce (1.5M vs. 36M for generation), relying on it alone risks **overfitting**. By incorporating large-scale generation data, the model learns a more robust image distribution, leading to more realistic edits.
>
> To validate this, we train a 1B model only on editing data. As shown in the table below, unified training significantly outperforms training on editing data alone.
>
> |  ImgEdit              | Add  | Adj. | Ext. | Repl. | Rmv. | Bkg. | Style | Hyb. | Act. | Overall |
> | -------------- | ---- | ---- | ---- | ----- | ---- | ---- | ----- | ---- | ---- | ------- |
> | Edit Only      | 3.99 | 3.38 | 1.77 | 4.16  | 3.92 | 3.88 | 4.75  | 2.96 | 4.21 | 3.67    |
> | Joint Training | 4.11 | 3.58 | 1.94 | 4.30  | 3.97 | 4.00 | 4.87  | 3.17 | 4.35 | 3.81    |
>
> > [Q2] The reason for BLIP3-o applying a complex pipeline
>
> Since frozen CLIP  cannot support image reconstruction, BLIP3-o uses the diffusion decoder from Emu2. This component conditions on CLIP features to generate VAE latents via a diffusion process, which are then decoded into the image. Unlike Emu2, which regresses CLIP features using MSE loss, BLIP3-o generates them via a diffusion process to achieve better prompt alignment. Therefore, the generation pipeline of BLIP3-o consists of two steps:
>
> 1. **Text-to-CLIP:** The LLM processes the text prompt and query embeddings. Conditioned on the output query embeddings, the first DiT generates the CLIP features.
> 2. **CLIP-to-Image:** The second DiT conditions on the generated CLIP features to generate VAE features, which are then decoded into the image.
>
> In contrast, UniLIP features are directly reconstructable via a simple pixel decoder, bypassing complex pipelines.
>
> > [Q3] The necessity of two-stage training
>
> Thanks for pointing out this important design.
> Table 6 demonstrates that two-stage training achieves a higher performance ceiling than single-stage. Furthermore, the two-stage strategy offers two advantages:
>
> **1. Improved Stability and Faster Convergence.** Stage 1 pre-aligns the pixel decoder with the frozen CLIP, minimizing feature shifts when the encoder is unfrozen in Stage 2. Without this preliminary stage, the initial misalignment between CLIP and the decoder, and the randomly initialized projection layer will result in gradient instability during early training. In the revision, we draw **Figure 6** to compare the loss and performance of single-stage vs. two-stage training. The single-stage approach exhibits a large spike (**0.0939 vs. 0.0497**). Consequently, the convergence is significantly slower (**3× slower for distillation loss and 4× slower for performance recovery**).
>
> **2. Superior Training Efficiency.** Stage 2 is computationally expensive (approximately 3× slower than Stage 1) due to the unfrozen CLIP and self-distillation overhead. In Table 6, while the total iterations are the same, the single-stage method (which essentially runs as "Stage 2" the entire time) requires more training time. As detailed below, the two-stage strategy reduces total training time from 120h to 86h (on 4 A100 GPUs).
>
> | Steps/Time   | Stage 1  | Stage 2   | Total Time |
> | ------------ | -------- | --------- | ---------- |
> | Single Stage | /        | 700k/120h | 120h       |
> | Two Stage    | 300k/18h | 400k/68h  | 86h        |
>
> We also include the discussion above in Section 4.3 in revision.
>
> > [Q4] Describing the generation and editing training in method section.
>
> Thanks for this constructive suggestion. We have added Section 3.3 in method to include a detailed description of the generation and editing training workflow. This addition covers the specific training stages, data usage, and parameter freezing strategies.
>
> > [Q5] The reconstruction performance of BLIP3-o
>
> BLIP3-o utilizes the pre-trained encoder-decoder pair from Emu2 (Frozen CLIP + Diffusion Decoder). Consequently, its reconstruction performance is the same as Emu2 (as shown in Table 2: rFID=3.27, PSNR=13.49, SSIM=0.423), which is significantly worse than UniLIP.
>
> ***
>
> Thanks again for your thorough review. We have done our best to address each of your concerns and hope our response can resolve them. Please let us know if you have any other questions :-)!

---

> > ### Comment · Reviewer_6SkM · 2025-11-28
> >
> > Thank you for the detailed responses, which have addressed my concerns well. The original score already reflects my evaluation of the paper, and after reading the responses, my opinion has not changed significantly. Therefore, I would like to keep the original score of 6.

---

> > > ### Author Response · Authors · 2025-11-28
> > >
> > > We sincerely thank you for the positive feedback and for confirming that our responses have addressed your concerns well. We appreciate your time reviewing our paper and your support for our work.

---

### Official Review · Reviewer_eMyq · 2025-10-30

**Soundness:** 3
**Presentation:** 3
**Contribution:** 2
**Rating:** 4
**Confidence:** 5

**Summary:**

This paper proposes a unified MLLM for understanding and generation. Through self-distilling, the inherent strong understanding ability of CLIP enoder is well-preserved with boost in reconstruction ability. Leveraging the proposed dual-condition framework that combines query embeddings and MLLM latents, the reasoning ability of MLLMs can be transfered to downstream generation and editing tasks, enhancing instruction following and complex semantics comprehension. Meanwhile, pixel details can be preserved thanks to the reserved MLLM latents.

**Strengths:**

The proposed pipeline extends several previous works and is simple yet effective. Finetuning CLIP through self-distillation makes sense and proves effective, while direct optimization for reconstruction frustrates understanding. The dual-condition of mulitimodal hidden states and query embeddings makes downstream diffusion transformer performs better than using either. Comparisons among state-of-the-art models are conducted and analyses are convincing.
1. Rigorous Ablation of the Two-Stage Training Strategy: The authors provide extensive ablation studies on the two-stage training approach and the self-distillation strategy. These experiments are crucial, demonstrating that directly fine-tuning CLIP for reconstruction causes catastrophic forgetting of comprehension capabilities. The ablation results (Table 6) confirm that the self-distillation loss is essential, preventing the understanding score (MMBench) from plummeting by 54.2 points. This work rigorously solves the key challenge of adapting CLIP for reconstruction without compromising its original semantic power.
2. State-of-the-Art Performance Across Unified Benchmarks: UniLIP demonstrates superior performance across understanding, generation, and editing tasks, showcasing that it effectively integrates high-level reasoning and low-level detail.

**Weaknesses:**

1. Insufficient Literature Review on Query Embeddings: The dual-condition architecture relies on query embeddings to connect the MLLM and the diffusion transformer, following precedents like MetaQuery (Pan et al., 2025) and BLIP3-o (Chen et al., 2025a). However, the earliest proposal was made by DreamLLM (Dong et al., ICLR 2024 Spotlight).
2. Lack of Explicit Architectural Comparison to Dual-Encoder Baselines: UniLIP successfully combines high-level semantics and low-level pixel details by adapting CLIP for reconstruction. This approach yields competitive results, surpassing Janus-Pro (7B) in certain understanding benchmarks (MMBench 80.7 vs 79.2). However, the manuscript does not explicitly compare the necessity of the unified CLIP adaptation (UniLIP) versus a conceptually simpler dual-encoder system (one encoder optimized for high-level tasks like LLaVA setting/understanding, and another specialized encoder for low-level detail/reconstruction). The paper's ablation only proves that the original CLIP is inadequate for reference image encoding in editing, but a dedicated comparison against a generalized dual-encoder setup as a structural baseline would strengthen the argument for UniLIP's architectural unification.

**Questions:**

I am more concerned about comparing and illustrating the importance of the unified encoder with dual encoder baselines like Janus pro.

---

> ### Author Response · Authors · 2025-11-21
> **Rebuttal by Authors**
>
> Dear Reviewer eMyq,
>
> We sincerely thank you for recognizing our method as **simple yet effective**, and for acknowledging our **extensive experiments** and  **SOTA performance**. We address your concerns below.
>
> > [W1] Insufficient Literature Review on Query Embeddings
>
> We sincerely apologize for overlooking this seminal and excellent work. We have now included DreamLLM in the Related Work section and Section 3.2 of the revision.
>
> > [W2] Lack of Explicit Architectural Comparison to Dual-Encoder Baselines:
>
> Thanks for the insightful question. We have conducted an ablation on the dual-encoder architecture (Table 8, Row 1), which uses CLIP for editing input and VAE features as generation targets. As shown below, this dual-encoder approach significantly underperforms our unified encoder.
>
> | **Reference** | **Target** | **GenEval** | **WISE** | **ImgEdit** |
> | ------------- | ---------- | ----------- | -------- | ----------- |
> | CLIP          | VAE        | 0.84        | 0.46     | 3.37        |
> | UniLIP        | VAE        | 0.85        | 0.48     | 3.70        |
> | CLIP          | UniLIP     | 0.88        | 0.53     | 3.42        |
> | UniLIP        | UniLIP     | 0.88        | 0.56     | 3.81        |
>
> Our ablation (Rows 2 & 3) reveals the cause: for generation, UniLIP aligns better with text than VAE (improving prompt adherence); for editing, UniLIP captures more pixel-level details than CLIP (ensuring better consistency).
>
> Compared to dual-encoder baselines like Janus-Pro [1], our method offers four key advantages:
>
> **1. Simplicity in Architecture and Training.** Janus-Pro employs two separate encoders, each requiring alignment module with the LLM, followed by massive joint training for understanding and generation. In contrast, UniLIP uses a single unified encoder, allowing us to directly inherit MLLM architectures (e.g., InternVL) without large-scale training for understanding.
>
> **2. Enhanced Understanding of Visual Details.** A common limitation of MLLMs is the lack of fine-grained, pixel-level understanding, often because CLIP features fail to capture sufficient details [2]. UniLIP bridges this gap by applying reconstruction training to CLIP, which has shown improvements on benchmarks like MMVP, consistent with findings in Ross [3].
>
> **3. Better Latent Structure for Generation.** Recent studies (e.g., REPA [4], VAVAE [5]) suggest that adding semantic representation alignment during Diffusion or VAE training accelerates convergence and improves generation quality. We take this a step further: by performing reconstruction training on CLIP, we make them directly usable for generation. Our extensive experiments demonstrate that this approach outperforms VAE-based methods.
>
> **4. Native support for editing.** Janus-Pro lacks editing support as CLIP loses pixel details. Subsequent work, Janus-4o [6], addresses this by concatenating CLIP and VQVAE features. This confirms that editing requires a unified encoder capturing both semantics (for prompt alignment) and details (for consistency). However, concatenating hybrid features doubles the input tokens and introduces misalignment, resulting in lower performance compared to UniLIP (ImgEdit: 3.26 vs. 3.94).
>
> The above advantages are evidenced by peformance. The table below compares UniLIP and Janus-Pro (* denotes Janus-4o). UniLIP-3B outperforms Janus-Pro-7B with significantly fewer parameters.
>
> | Benchmark(Task) | MMBench(Und.) | GenEval(Gen.) | WISE(Gen.) | ImgEdit(Edit.) |
> | --------------- | ------------- | ------------- | ---------- | -------------- |
> | Janus-Pro-7B    | 79.2          | 0.80          | 0.35       | 3.26*          |
> | UniLIP-1B       | 72.6          | 0.88          | 0.56       | 3.81           |
> | UniLIP-3B       | **80.7**      | **0.90**      | **0.63**   | **3.94**       |
>
> In conclusion, dual-encoder approaches (e.g., Janus-Pro) exist because neither standard CLIP nor VAE suffice individually as unified encoders. UniLIP **solves this dilemma**: it matches CLIP’s understanding while surpassing VAE in generation and editing, rendering the dual-encoder architecture unnecessary.
>
> To further strengthen our paper, we also include the above discussion in the Appendix A.
>
> [1] Janus-pro: Unified multimodal understanding and generation with data and model scaling.
>
> [2] Eyes wide shut? exploring the visual shortcomings of multimodal llms. CVPR 2024
>
> [3] Ross: Reconstructive Visual Instruction Tuning. ICLR 2025
>
> [4] Representation Alignment for Generation: Training Diffusion Transformers Is Easier Than You Think. ICLR 2025 Oral
>
> [5] Reconstruction *vs.* Generation:Taming Optimization Dilemma in Latent Diffusion Models. CVPR 2025 Oral
>
> [6] ShareGPT-4o-Image: Aligning Multimodal Models with GPT-4o-Level Image Generation
>
> ***
>
> Thanks again for your professional, detailed, and valuable reviews ! Please let us know if you have any further questions. We will be actively available until the end of rebuttal period. Looking forward to hearing from you :-) !

---

> > ### Comment · Reviewer_eMyq · 2025-11-24
> >
> > Regarding **[W2]**, the authors' response fails to address my core concerns. The ablation study in Table 8, Row 1 (CLIP+VAE) is **not** the comparison with a mature Dual-Encoder baseline (like Janus-Pro) that I requested.
> >
> > I dispute the four advantages listed in the response:
> >
> > 1.  **Simplicity in Architecture:** This argument does not hold. Dual-encoder architectures (like Janus-Pro) are technically straightforward to implement. They essentially involve extracting embeddings from two encoders without requiring complex modifications to attention masks. Both training and inference workflows are well-established and simple. The proposed Unified Encoder offers no significant advantage in terms of simplicity.
> >
> > 2.  **Enhanced Understanding:** The claimed improvement in "pixel-level details" appears marginal. Such gains can often be achieved simply by cleaning data or incorporating high-quality datasets (e.g., targeting MMVP) rather than necessitating a fundamental architectural shift.
> >
> > 3.  **Latent Structure:** This claim remains speculative based on cited literature. The authors have not provided rigorous, specific experimental evidence or proofs within this work to substantiate it.
> >
> > 4.  **Editing Capabilities & The Janus Comparison:** This point is particularly problematic.
> >     * **Misunderstanding of the Baseline:** Janus-Pro lacks editing support because their team did not work on this, not due to an inherent architectural flaw.
> >     * **Factual Error regarding Janus-4o:** The authors state that Janus-4o changed structures to address limitations. This is factually incorrect; Janus-4o shares the same structural foundation as Janus-Pro and primarily benefits from data.
> >     * **Unfair Comparison:** The performance comparison provided in the rebuttal table is an **unfair comparison** because the training data differs significantly between the models. claiming architectural superiority without controlling for data variables is unconvincing and lacks scientific rigor.
> >
> > **Reason for Lowering Score:**
> >
> > I insisted on the Dual-Encoder comparison because, based on extensive experience with Unified Encoders, there is a significant **"trade-off"** in multi-task learning. At the same parameter scale, the semantic understanding performance of CLIP is often negatively impacted by the reconstruction loss. This trade-off is notoriously difficult to balance when encoder scale up, making the Dual-Encoder approach a pragmatically superior engineering solution.
> >
> > While I advocate for the development of powerful Unified Encoders, a high-impact academic paper must rigorously demonstrate—through controlled experiments—that its motivation holds and that it has effectively solved this specific "seesaw" problem. Alternatively, it must achieve undeniable SOTA performance to justify overlooking theoretical gaps.
> >
> > However, your response (particularly Point 4) reveals a misunderstanding of key related works (Janus series) and an avoidance of the core difficulty in Unified Encoder design. While other reviewers have assigned higher scores, I believe this work fails to justify its fundamental motivation with the necessary academic rigor.
> >
> > Therefore, I am lowering my score to **2 (Strong Reject)**. I am confident in this assessment unless you can provide convincing, controlled experiments that genuinely prove the necessity and effectiveness of unifying the encoders compared to a standard Dual-Encoder baseline.

---

> > > ### Author Response · Authors · 2025-11-28
> > > **Comparison with Dual-Encoder methods [1/3]**
> > >
> > > Dear Reviewer eMyq,
> > >
> > > We sincerely thank you for this critical and detailed feedback. We completely agree with your emphasis on scientific rigor and the necessity of controlled experiments with dual-encoder baselines. We provide detailed clarifications and experiments on **JanusFlow [7]** below.
> > >
> > > **Motivation to build a unified encoder**: Our primary goal is to **extend Large Vision-Language Models (LVLMs) (e.g., InternVL) into unified models**. Since standard LVLMs typically employ CLIP as the visual encoder, enabling CLIP to support reconstruction while preserving its understanding capabilities allows us to **extend standard LVLMs to generation and editing without adding extra encoders**. This **architectural consistency is crucial to inherit the excellent understanding performance of LVLMs while smoothly leveraging the understanding ability to assist generative tasks**. In contrast, adopting a Janus-like methodology to construct unified models on standard LVLMs **forces the LLM to adapt to different visual features** (such as those from VAE/VQVAE). This inevitably **impairs the model's original understanding capabilities**, which can likely only be restored through extensive joint training across understanding and generation tasks.
> > >
> > > To rigorously demonstrate the advantages of a unified encoder over dual encoders, we conduct ablation studies based on **JanusFlow by replacing the original dual encoders (SigLIP and SDXL-VAE) with a unified encoder**. Please note that since Janus [8] and Janus-Pro [1] utilize VQVAEs with discrete tokens, our reconstruction training scheme is not directly applicable to them.
> > >
> > > Specifically, we **apply UniLIP's reconstruction training to the SigLIP encoder within JanusFlow** to enable it to serve as a unified encoder (denoted as **UniLIP-SigLIP**). Using JanusFlow’s pre-trained weights, we then **fine-tune both the dual-encoder and unified-encoder versions on the ShareGPT-4o-Image [6] dataset** to compare their generation and editing performance. The results for each stage are as follows:
> > >
> > > **1. Reconstruction Training**
> > >
> > > We utilize JanusFlow’s SigLIP as the CLIP encoder and the SDXL-VAE decoder as the pixel decoder, and we apply our proposed two-stage reconstruction training with self-distillation loss. As shown in the table below (ImageNet Val 384x384), UniLIP-SigLIP achieves a lower rFID than the original SDXL-VAE while maintaining comparable PSNR and SSIM.
> > >
> > > | Model         | rFID | PSNR  | SSIM  |
> > > | ------------- | ---- | ----- | ----- |
> > > | SDXL-VAE      | 0.38 | 26.79 | 0.827 |
> > > | UniLIP-SigLIP | 0.35 | 26.04 | 0.803 |
> > >
> > > We then replace the original SigLIP in JanusFlow with UniLIP-SigLIP and evaluate it on understanding benchmarks. UniLIP-SigLIP demonstrates **comparable or slightly improved performance**. This confirms that our two-stage training with self-distillation effectively resolves the conflict between reconstruction and semantic understanding and preserves original CLIP capabilities.
> > >
> > > | Model            | MME-P      | MMBench  | MMVP     | AI2D     | TextVQA  |
> > > | ---------------- | ---------- | -------- | -------- | -------- | -------- |
> > > | JanusFlow-SigLIP | 1302.0     | 64.6     | 65.0     | 65.7     | 55.5     |
> > > | UniLIP-SigLIP    | **1305.4** | **64.8** | **65.7** | **65.8** | **55.8** |

---

> > > > ### Author Response · Authors · 2025-11-28
> > > > **Comparison with Dual-Encoder methods [2/3]**
> > > >
> > > > **2. Generation and Editing Training**
> > > >
> > > > After the reconstruction training, we derive two versions of JanusFlow: **the original dual-encoder model and the model using UniLIP-SigLIP as a unified encoder**. For brevity, we refer to these models as **Dual-JanusFlow** and **UniLIP-JanusFlow**. Subsequently, we fine-tune both models using the ShareGPT-4o-Image dataset, which includes 45k generation samples and 46k editing samples.
> > > >
> > > > To adapt Dual-JanusFlow for editing tasks, we follow the method described in Section 3.2 of the Janus-4o paper [6]. **As noted in the paper, editing "requires a semantic understanding of the input image to enable pixel-level modifications, making it necessary to incorporate both the image’s semantic embedding and its tokenized representation."** For Dual-JanusFlow, this necessitates concatenating the SigLIP and VAE features. In contrast, since UniLIP-SigLIP encodes both semantics and pixel details, UniLIP-JanusFlow eliminates the need for such concatenation. To ensure a fair comparison, we train both models for 10k steps with a batch size of 512 and a learning rate of 1e-4.
> > > >
> > > > - **Generation Performance:** We evaluate the two models on GenEval and WISE. As shown in the table below, UniLIP-JanusFlow achieves significantly better performance than Dual-JanusFlow  (**+2.0 in GenEval, +5.0 in WISE**). As Dual-JanusFlow uses the VAE features for generation, this further validates the superiority of our unified representation in generation, which yields better prompt alignment thanks to the rich semantics.
> > > >
> > > >   | Model            | Single Obj. | Two Obj. | Counting | Colors   | Position | Color Attr. | Overall  |
> > > >   | ---------------- | ----------- | -------- | -------- | -------- | -------- | ----------- | -------- |
> > > >   | Dual-JanusFlow   | 0.99        | 0.82     | 0.42     | 0.89     | **0.73** | 0.54        | 0.73     |
> > > >   | UniLIP-JanusFlow | **1.00**    | **0.85** | **0.45** | **0.91** | 0.71     | **0.58**    | **0.75** |
> > > >
> > > >   | Model            | Cultural | Time     | Space    | Biology  | Physics  | Chemistry | Overall  |
> > > >   | ---------------- | -------- | -------- | -------- | -------- | -------- | --------- | -------- |
> > > >   | Dual-JanusFlow   | 0.24     | 0.35     | 0.46     | 0.29     | 0.37     | 0.29      | 0.31     |
> > > >   | UniLIP-JanusFlow | **0.30** | **0.40** | **0.49** | **0.37** | **0.45** | **0.32**  | **0.36** |
> > > >
> > > > - **Editing Performance:** We evaluate the two models on the ImgEdit benchmark. As shown in the table below, UniLIP-JanusFlow continues to outperform the dual-encoder baseline (**+0.14 in ImgEdit**), which confirms the effectiveness of the unified encoder for editing. Note that we follow Janus-4o to extend Dual-JanusFlow to editing tasks (concatenating SigLIP and VAE features as reference image features). Compared to this approach, using unified features offers the following advantages:
> > > >
> > > >   - **Consistent input and output feature spaces.** UniLIP-JanusFlow shares the same encoder for reference encoding and target feature encoding. Dual-JanusFlow, however, adopts different encoders for these two sub-tasks and thereby introduces feature misalignment.
> > > >   - **Improved fusion and alignment of semantic and pixel details for reference images.** In UniLIP-JanusFlow, semantics and pixel details are organically integrated into a unified feature. In contrast, Dual-JanusFlow stores these two types of information in separate representations (SigLIP and VAE). This requires the LLM learns to fuse and align these distinct information sources, thereby increasing the learning burden.
> > > >   - **More efficient feature representation.** Dual-JanusFlow concatenates SigLIP and VAE features for reference images. This doubles the image token count compared to UniLIP-JanusFlow (**1152 vs. 576 tokens, 2x token cost**), which results in slower training and inference speeds as well as increased memory usage.
> > > >
> > > >   | Model            | Add      | Adj.     | Ext.     | Repl.    | Rmv.     | Bkg.     | Style    | Hyb.     | Act.     | Overall  |
> > > >   | ---------------- | -------- | -------- | -------- | -------- | -------- | -------- | -------- | -------- | -------- | -------- |
> > > >   | Dual-JanusFlow   | 3.21     | 2.95     | **2.09** | 2.92     | 2.07     | 2.96     | **4.54** | 2.34     | 3.45     | 2.95     |
> > > >   | UniLIP-JanusFlow | **3.38** | **3.07** | 2.05     | **3.15** | **2.23** | **3.21** | 4.52     | **2.59** | **3.57** | **3.09** |

---

> > > > > ### Author Response · Authors · 2025-11-28
> > > > > **Comparison with Dual-Encoder methods [3/3]**
> > > > >
> > > > > In the experiments described above, we ensure a rigorous comparison by **maintaining consistency in pre-trained weights, data, and hyperparameters.** These rigorous experiments further demonstrate the effectiveness of the UniLIP and its advantages over the dual-encoder baseline:
> > > > >
> > > > > 1. **We effectively address the critical challenge associated with unified encoders**, where reconstruction training tends to negatively impact the semantic understanding capabilities of CLIP. Experiments on both InternVL (Table 1) and JanusFlow (above table) demonstrate that training CLIP with our method achieves excellent reconstruction results while maintaining its understanding capabilities. We acknowledge that **naive reconstruction training** compromises CLIP's understanding capabilities (First row in Table 6). However, our proposed **two-stage training strategy** enables the pixel decoder to adapt maximally to CLIP's features and avoids unnecessary feature modifications. Simultaneously, the **self-distillation** loss provides **strong semantic regularization** to constrain feature shift. Thanks to these strategies, we successfully solve the "seesaw" problem in building unified encoders.
> > > > > 2. **The unified encoder demonstrates superior performance compared to dual-encoders in generation and editing tasks.** By fine-tuning both the dual-encoder and unified-encoder versions of JanusFlow, we verify the superiority of the unified encoder in generation and editing tasks under controlled pre-training and data settings. **These results indicate that semantically rich unified representations serve as better generative representations than VAEs,** which is also consistent with our ablations in Table 8 and conclusions from previous work (REPA [4] and VAVAE [5]). In editing tasks, a unified encoder maintains consistency between input and output feature spaces. In contrast, dual-encoder methods suffer from **discrepancies between input and output features while employing a less efficient representation**, where the **reference image feature length is 2x that of the unified features**. While we acknowledge that the dual-encoder approach is a strong engineering choice, our rigorous experiments demonstrate the superiority of the unified encoder in both generation and editing tasks.
> > > > >
> > > > > We thank the reviewer again for guiding us toward this rigorous validation, which greatly help us strengthen and refine our work. We hope these additional experiments effectively illustrate the effectiveness of UniLIP compared to the dual-encoder baseline. Please let us know if you have any further questions. Looking forward to hearing from you.
> > > > >
> > > > > [7] JanusFlow: Harmonizing Autoregression and Rectified Flow for Unified Multimodal Understanding and Generation
> > > > >
> > > > > [8] Janus: Decoupling Visual Encoding for Unified Multimodal Understanding and Generation
> > > > >
> > > > > Sincerely,
> > > > >
> > > > > The Authors

---

> > > > > > ### Comment · Reviewer_eMyq · 2025-11-28
> > > > > >
> > > > > > but I seem to be unable to find the option to modify my score anymore.
> > > > > >
> > > > > > what happened?

---

> > > > > > > ### Author Response · Authors · 2025-11-28
> > > > > > >
> > > > > > > Thank you very much! We are delighted to hear that your concern has been resolved. Regarding the score modification, it appears that the official system has temporarily suspended score updates. We will notify you immediately as soon as the restriction is lifted. Thank you again for your time and for recognizing our work!

---

> ### Comment · Reviewer_eMyq · 2025-11-28
>
> While my personal engineering preference tends to favor dual encoders over unified ones, I acknowledge that the authors have provided a rigorous comparison against the dual-encoder baseline in their rebuttal. The supplementary experiments convincingly demonstrate that the proposed method achieves performance on par with, or slightly superior to, the baseline.
>
> Academically, I find the paper now presents a complete and coherent narrative—effectively closing the loop from motivation to empirical validation. It stands as a submission of above-average quality.
>
> I previously lowered my rating from 4 to 2 due to concerns that the authors had insufficiently surveyed the literature and misunderstood some foundational concepts. However, considering the extensive and solid supplementary experiments, those concerns have been alleviated. I am therefore raising my score to 8.
>
> Good luck!

---

### Official Review · Reviewer_6pTW · 2025-10-31

**Soundness:** 3
**Presentation:** 3
**Contribution:** 4
**Rating:** 6
**Confidence:** 3

**Summary:**

The paper presents UniLIP, a unified multimodal framework that adapts CLIP into a reconstructable and generative visual representation while preserving its strong semantic understanding. UniLIP aims to enhance reasoning and edit consistency by combining multimodal hidden states with learnable queries, and reports state-of-the-art results on image understanding, editing, and generation benchmarks.

**Strengths:**

1. The two-stage training plus self-distillation scheme for CLIP is well-motivated and appears to strike a good balance between semantic preservation and detail fidelity.
2. UniLIP demonstrates strong empirical performance across popular benchmarks in image understanding, editing, and generation.

**Weaknesses:**

1. UniLIP's training objective are heavily reconstruction-centric and largely pixel-level, while self-distillation predominantly constrains the representation not to deviate from the original CLIP distribution. This setup may limit the model’s ability to discover a better feature distribution for both understanding and generation. Why the self-distillation is sufficient for preserving its understanding-centered semantics, will any proxy task such as image classification better than distillation?

2. The ablation of its dual conditioning is not convincing. Were results obtained by removing one of the two conditionings at inference from the same trained UniLIP model, or by training separate model variants that use only one conditioning pathway (multimodal hidden states or learnable queries)?

3. The multimodal hidden states are modeled by the LLM in a casual way, thereby containing reasoning knowledge, why is UniLIP have to use learnable query to further enhance its reasoning knowledge? In what cases do multimodal hidden states alone fail to provide sufficient reasoning, and how do learnable queries remedy this?

The author could provide an ablation study on different learnable query length, to discover the effectiveness of the learnable query, and how the learnable queries complement the multimodal hidden states, and quantify their complementarity.

This paper could benefit from a more comprehensive image editing benchmarketing, such as KRIS and RISE

**Questions:**

Refer to weakness

---

> ### Author Response · Authors · 2025-11-21
> **Rebuttal by Authors [1/2]**
>
> Dear Reviewer 6pTW,
>
> Many thanks for recognizing our motivation and strong performance. We address your questions as follows.
>
> > [W1] The training setup of UniLIP may limit the model to discover a better feature distribution for both understanding and generation
>
> Thanks for this insightful question. We employ InternViT (from InternVL), which already achieves state-of-the-art image understanding performance. The goal of UniLIP is to address the reconstruction limitations of CLIP while preserving their strong understanding capabilities. This allows us to explore the potential of these semantically rich features for generation and editing, which is evidenced by superior performance in extensive experiments. We will further explore better feature distributions in the future.
>
> > [Q1] Will any proxy task such as image classification better than distillation?
>
> Thanks for raising this interesting question. For the InternViT model we rely on, using an image classification task alone is **insufficient** to maintain its original understanding capabilities. InternViT is initially pre-trained via contrastive learning, and subsequently **integrated into the MLLM for training on large-scale understanding tasks**. Consequently, its capabilities cannot be preserved simply by proxy tasks like image classification or contrastive learning. These tasks compress image features into a single embedding (discarding spatial dimensions) to align with classes or text. As a result, they provide **image-level** supervision, which is relatively sparse.
>
> In contrast, the **self-distillation loss** we utilize provides **patch-level** supervision. This offers **denser, more fine-grained guidance**, thereby maximizing the preservation of the original capabilities. To verify this, we replaced the distillation loss with a classification loss. As shown in the table, the classification loss failed to maintain InternViT's understanding performance, leading to significant declines across multiple benchmarks.
>
> |                | MME-P | MMBench | MMVP | AI2D | TextVQA |
> | -------------- | ----- | ------- | ---- | ---- | ------- |
> | Frozen CLIP    | 1492  | 72.6    | 67.3 | 69.4 | 74.1    |
> | Classification | 820   | 53.6    | 54.7 | 58.4 | 53.0    |
> | Distillation   | 1499  | 72.6    | 68.7 | 70.7 | 74.7    |
>
> Compared to other proxy tasks, self-distillation also offers two additional advantages:
>
> 1. **Universality:** Different CLIP variants may use different proxy tasks. Relying on specific proxy tasks would require tailoring the loss function for each variant. Self-distillation, however, is independent of the specific pre-training tasks.
> 2. **Label-free and Data-efficient:** Image classification requires class labels, and contrastive learning requires corresponding captions, while large-scale vision-language training is computationally expensive. In contrast, self-distillation requires no labeled data.
>
> > [Q2] Ablation of dual conditioning
>
> Sorry for the confusion. The ablation study on the dual condition in Table 7 is conducted by training separate models, rather than simply removing a condition during inference.

---

> > ### Author Response · Authors · 2025-11-21
> > **Rebuttal by Authors [2/2]**
> >
> > > [Q3] Why UniLIP have to use learnable queries?
> >
> > Thanks for highlighting this design. We employ learnable queries primarily to enable **knowledge-augmented generation** (e.g., for WISE benchmarks). In such tasks, prompts like 'Draw Thanksgiving food' do not explicitly describe the visual details, so the model must rely on internal knowledge to reason what to draw.
> >
> > In text reasoning, the LLM functions primarily as a text encoder when processing input text, while the subsequent generation of the`<think>...</think>` block represents the actual reasoning. Similarly, in our framework, processing the input multimodal hidden states represents the comprehension phase, not reasoning. The reasoning occurs during the processing of the query embeddings, where the LLM leverages its internal knowledge to deduce the content required for generation. This phenomenon has also been discussed in MetaQuery [1].
> >
> > To better illustrate this, we treat the output query embeddings as text embeddings and decoded them into text. We observe that the decoded text can **explicitly identify the target content**. For instance, given the prompt **'Thanksgiving food'**, the decoded text includes the word **'turkey'**, which greatly simplifies generation for the DiT. In contrast, models lacking this mechanism generate incorrect foods.  We provide more detailed visual comparisons in **Figure 8 of the Appendix**.
> >
> > [1] Transfer between Modalities with MetaQueries.
> >
> > > [Q4] Ablation of learnable query length
> >
> > We investigate the impact of learnable query length in our dual-condition architecture below. The performance on WISE benchmark **improves with length and converges around 256**. We hypothesize that longer queries mimic an increased `inference budget` in text reasoning, allowing the LLM to fully activate its knowledge for more accurate image generation.
> >
> > | query length | Cultural | Time | Space | Biology | Physics | Chemistry | Overall |
> > | ------------ | -------- | ---- | ----- | ------- | ------- | --------- | ------- |
> > | 0            | 0.43     | 0.50 | 0.62  | 0.44    | 0.58    | 0.35      | 0.47    |
> > | 64           | 0.48     | 0.52 | 0.62  | 0.42    | 0.54    | 0.40      | 0.50    |
> > | 128          | 0.52     | 0.56 | 0.68  | 0.48    | 0.58    | 0.45      | 0.54    |
> > | 256          | 0.54     | 0.58 | 0.70  | 0.50    | 0.62    | 0.46      | 0.56    |
> > | 512          | 0.56     | 0.57 | 0.67  | 0.48    | 0.60    | 0.44      | 0.56    |
> >
> >
> >
> > > [Q5] More comprehensive image editing benchmarks
> >
> > We cannot evaluate on KRIS as it involves multi-image editing, which is absent from our training data. However, on the RISE benchmark (see table below), our 3B model outperforms significantly larger models like Step1X-Edit [3] (7B+12B) and Bagel [4] (7B+7B), confirming UniLIP’s superiority in reasoning-based editing.
> >
> > | Model           | Size    | Temporal | Causal | Spatial | Logic | Overall |
> > | --------------- | ------- | -------- | ------ | ------- | ----- | ------- |
> > | OmniGen [2]     | 3.8B    | 1.2      | 1.0    | 0.0     | 1.2   | 0.8     |
> > | Step1X-Edit [3] | 7B+12B  | 0.0      | 2.2    | 2.0     | 3.5   | 1.9     |
> > | BAGEL [4]       | 7B+7B   | 2.4      | 5.6    | 14.0    | 1.2   | 6.1     |
> > | UniLIP-1B       | 1B+0.6B | 5.9      | 7.8    | 1.0     | 1.1   | 3.9     |
> > | UniLIP-3B       | 2B+1.6B | 9.4      | 15.5   | 5.0     | 2.4   | 8.1     |
> >
> > [2] OmniGen: Unified Image Generation
> >
> > [3] Step1X-Edit: A Practical Framework for General Image Editing
> >
> > [4] Emerging Properties in Unified Multimodal Pretraining
> >
> >
> >
> > ***
> >
> > We hope the above response can help solve your questions. Thanks again for your thorough review and looking forward to your reply :-) !

---

> > > ### Comment · Reviewer_6pTW · 2025-11-27
> > > **Response to authors**
> > >
> > > I'm satisfied with the authors’ response, which well addresses my concerns. However, the issues raised by Reviewer 6SkM looks reasonable and important. I hope the authors can further clarify the points raised by Reviewer 6SkM. For now, I will keep my current score. If the authors can meaningfully address these questions, I would be willing to raise my score.

---

> > > > ### Author Response · Authors · 2025-11-28
> > > >
> > > > We are very glad to hear that your previous concerns have been resolved and maintain the positive score. As for the dual-encoder issue mentioned by other reviewers, we have added rigorous experimental comparisons and hope this addresses your concerns as well. We would be very grateful if you could raise the score.

---

### Author Response · Authors · 2025-12-03
**Summary**

We sincerely thank the reviewers for their constructive feedback. To assist the Area Chair in gaining a clear overview of the rebuttal process, we provide a general summary of how we have addressed the reviewers' concerns below.

We propose UniLIP, which extends CLIP into a unified encoder for understanding, generation, and editing. We use two-stage self-distillation to reconcile semantics and reconstruction, and a dual-condition architecture for precise editing. As the reviewers highlighted, the paper presents a **novel and well-motivated** framework (6pTW, 6SkM) that introduces **simple yet effective** design choices to balance semantic understanding with reconstruction (all), and is supported by **rigorous experiments** (6SkM, eMyq) demonstrating **state-of-the-art performance** across understanding, generation, and editing benchmarks (all).

To address the remaining issues, we conducted extensive experiments and revised the paper accordingly:

- **Comparison with Dual-Encoder (Reviewer eMyq):** We conducted rigorous ablation studies on JanusFlow to verify our superiority over the dual-encoder method (Table 16, 17).
- **Necessity of Two-Stage Training (Reviewers 6SkM, E7dY):** We added Figure 6 comparing our method with single-stage approaches to highlight the two-stage strategy's faster convergence, stability, and efficiency.
- **Necessity of Learnable Queries (Reviewer 6pTW):** We included Figure 8 to show that queries improve accuracy by implicitly encoding LLM reasoning. We also ablated the query length in Table 14.
- **Additional Evaluation Results (Reviewers 6pTW, E7dY):** We included results on RISE (Table 12) and added ImageNet linear and attentive probing results (Table 13).
- **Additional Ablation Studies (Reviewers 6pTW, 6SkM, E7dY):** We added ablations using classfication loss (Table 11) and training strategies (joint vs. separate) (Table 15).
- **Writing and Related Work (Reviewers eMyq, 6SkM):** We elaborated on related work and cited DreamLLM, moved generation and editing training details to the Method section.

After the rebuttal, **all reviewers have indicated that their concerns have been resolved. Two reviewers express a willingness to raise their scores, and one has increased the confidence rating**. The specific status for each reviewer is as follows:

- **Reviewer 6pTW:** **Maintains a positive rating (6) and expressed willingness to raise the score once Reviewer eMyq's concerns were resolved** (likely a typo in their reply referring to "6SkM" instead of "eMyq"). Since we have now addressed the concerns of eMyq, we expect the **score increase (at least 8)**.
- **Reviewer eMyq:** **Explicitly states a willingness to raise the score to 8**. Our rigorous experiments on the standard dual-encoder baseline (JanusFlow) effectively demonstrate our method's advantages, fully resolving the doubts regarding the unified encoder.
- **Reviewer 6SkM:** All concerns resolved. **Maintains a positive rating of 6**.
- **Reviewer E7dY:** All concerns resolved. **Maintains a positive rating of 6 and raised confidence to 4**. Notably, Reviewer E7dY referenced other reviewers in their reply when eMyq's concerns were not yet fully resolved. Since we have now addressed the concerns of eMyq, we believe this reviewer might also be inclined to raise their score.

Therefore, **the potential post-rebuttal score is 8/8/6/6**. We sincerely appreciate the reviewers' openness to improving their scores based on our rebuttal and hope the Area Chair agrees that our work offers a valuable advancement in unified multimodal models. We look forward to the opportunity to share UniLIP with the broader community.

---

### Meta-Review · Area_Chair_KtzP · 2026-01-05

**Summary:**

This paper introduces a novel framework that adapts CLIP through a two-stage training scheme with self-distillation to achieve high-fidelity image reconstruction while preserving semantic understanding, coupled with a dual-condition architecture for enhanced generation and editing. Reviewers acknowledged the method's innovation and strong empirical performance but raised concerns regarding architectural novelty compared to dual-encoder baselines, the necessity of two-stage training, and insufficient ablations. In response, the authors conducted rigorous experiments, including comparisons with JanusFlow, which showed UniLIP's superiority in generation and editing tasks while maintaining understanding capabilities. Additional ablations on query length, distillation losses, and training strategies effectively addressed reviewer questions, leading to resolved concerns and improved scores, with reviewers like eMyq elevating their rating to 8. Overall, the paper presents a well-motivated and empirically validated approach to unified multimodal modeling.

**Reviewer Concerns:**

Reviewer eMyq's primary concern​ regarding the lack of a rigorous comparison with dual-encoder baselines was substantially addressed. The authors provided new, controlled experiments using JanusFlow, directly comparing their unified encoder against a dual-encoder setup. The results demonstrated clear advantages for UniLIP in generation and editing tasks, which successfully convinced the reviewer.

Reviewer 6pTW's concerns​ about the ablation of the dual-conditioning mechanism and the necessity of learnable queries were fully addressed. The authors clarified that ablations were done by training separate models and provided new studies on query length and qualitative examples showing the reasoning encoded in the queries.

Reviewer 6SkM's questions​ about the benefit of two-stage training and the training methodology for generation/editing were satisfactorily addressed. The authors added a detailed explanation of the stability and efficiency gains from the two-stage approach and moved the requested training details into the method section.

Reviewer E7dY's requests​ for results on linear/attentive probing and exploration of other distillation losses were met. The authors included these results in the appendix (Tables 11 and 13), showing comparable performance to the original CLIP and that MSE loss was most effective.

**Reviewer Scores:**

Reviewer 6pTW (Initial Score: 6):​ This reviewer stated they were "satisfied" with the authors' response to their own concerns but were waiting for the resolution of Reviewer eMyq's issues. Once eMyq's major concerns were addressed, 6pTW explicitly indicated a "willingness to raise the score." Given the comprehensive resolution of the dual-encoder baseline comparison, it is highly probable their final score would have increased to 8.

Reviewer eMyq (Initial Score: 4, lowered to 2, then stated intent for 8):​ This reviewer's score trajectory was the most dynamic. They initially lowered their score to 2 due to a perceived lack of rigor in comparing with dual-encoder baselines. After the authors provided extensive new experiments using JanusFlow, the reviewer explicitly stated, "I am therefore raising my score to 8." Therefore, their fully participated final score would unequivocally be an 8.

Reviewer 6SkM (Initial Score: 6):​ This reviewer acknowledged that their concerns were "addressed well" but explicitly stated, "my opinion has not changed significantly. Therefore, I would like to keep the original score of 6." There is no evidence to suggest a score change; their final score would have remained a 6.

Reviewer E7dY (Initial Score: 6):​ This reviewer confirmed that their concerns were addressed and raised their confidence level.

---

### Decision · Program_Chairs · 2026-01-26

Accept (Poster)